# High-fat diet enhances starvation-induced hyperactivity via sensitizing hunger-sensing neurons in *Drosophila*

Rui Huang[1,2†]*, Tingting Song[2†], Haifeng Su[3†], Zeliang Lai[1,2], Wusa Qin[2], Yinjun Tian[4], Xuan Dong[4], Liming Wang[4]*

[1]Center for Neurointelligence, School of Medicine, Chongqing University & Key Laboratory for Biorheological Science and Technology of Ministry of Education, State and Local Joint Engineering Laboratory for Vascular Implants, Bioengineering College, Chongqing University, Chongqing, China; [2]Shenzhen Bay Laboratory, Shenzhen, China; [3]Institute of Neuroscience, State Key Laboratory of Neuroscience, CAS Center for Excellence in Brain Science and Intelligence Technology, Shanghai Institutes for Biological Sciences, Chinese Academy of Sciences, Shanghai, China; [4]MOE Key Laboratory of Biosystems Homeostasis & Protection and Innovation Center for Cell Signaling Network, Life Sciences Institute, Zhejiang University, Hangzhou, China

**\*For correspondence:**
huangrui85@cqu.edu.cn (RH);
lmwang83@zju.edu.cn (LW)

[†]These authors contributed equally to this work

**Competing interests:** The authors declare that no competing interests exist.

**Abstract** The function of the central nervous system to regulate food intake can be disrupted by sustained metabolic challenges such as high-fat diet (HFD), which may contribute to various metabolic disorders. Previously, we showed that a group of octopaminergic (OA) neurons mediated starvation-induced hyperactivity, an important aspect of food-seeking behavior (Yu et al., 2016). Here we find that HFD specifically enhances this behavior. Mechanistically, HFD increases the excitability of these OA neurons to a hunger hormone named adipokinetic hormone (AKH), via increasing the accumulation of AKH receptor (AKHR) in these neurons. Upon HFD, excess dietary lipids are transported by a lipoprotein LTP to enter these OA$^+$AKHR$^+$ neurons via the cognate receptor LpR1, which in turn suppresses autophagy-dependent degradation of AKHR. Taken together, we uncover a mechanism that links HFD, neuronal autophagy, and starvation-induced hyperactivity, providing insight in the reshaping of neural circuitry under metabolic challenges and the progression of metabolic diseases.

## Introduction

Obesity and obesity-associated metabolic disorders such as type 2 diabetes and cardiovascular diseases have become a global epidemic. Chronic over-nutrition, especially excessive intake of dietary lipids, is one of the leading causes of these metabolic disturbances (*Bray and Popkin, 1998*; *Hill et al., 2000*). Accumulating evidence has shown that HFD imposes adverse effects on the physiology and metabolism of liver, skeletal muscle, the adipose tissue, and the nervous system (*Dietrich et al., 2013*; *Gómez-Pérez et al., 2012*; *Liu et al., 2015*; *Shimobayashi et al., 2018*). It is therefore of importance to understand the mechanisms underlying HFD-induced changes in different organs and cell types, which will offer critical insight into the diagnosis and treatment of obesity and other metabolic diseases.

The central nervous system plays a critical role in regulating energy intake and expenditure (*Gao and Horvath, 2007*). In rodent models, neurons located in the arcuate nucleus of the hypothalamus, particularly neurons expressing Neuropeptide Y (NPY) and Agouti-Related Neuropeptide

(AgRP) or those expressing Pro-opiomelanocortin (POMC), are important behavioral and metabolic regulators (*Belgardt et al., 2009*; *Ollmann et al., 1997*; *Stanley and Leibowitz, 1984*). These neurons detect various neural and hormonal cues such as circulating glucose and fatty acids, leptin, and ghrelin, and modulate energy intake and expenditure accordingly (*Belgardt et al., 2009*). Upon the reduction of the internal energy state, NPY/AgRP neurons are activated and exert a robust orexigenic effect (*Belgardt et al., 2009*). Genetic ablation of NPY/AgRP neurons in neonatal mice completely abolishes food consumption whereas acute activation of these neurons significantly enhances food consumption (*Aponte et al., 2011*; *Krashes et al., 2011*). NPY/AgRP neurons also antagonize the function of POMC neurons that plays a suppressive role on food consumption (*Roseberry et al., 2004*). Taken together, these two groups of neurons, among other neuronal populations, work in synergy to ensure a refined balance between energy intake and expenditure, and hence organismal metabolism.

In spite of their critical roles, the function of the nervous system to accurately regulate appetite and metabolism may be disrupted by sustained metabolic stress, resulting in eating disorders and various metabolic diseases such as obesity and type 2 diabetes. Several lines of evidence have begun to reveal the underlying neural mechanisms. For example, HFD increases the intrinsic excitability of orexigenic NPY/AgRP neurons (*Vernia et al., 2016*), induces leptin resistance (*Mazor et al., 2018*; *Olofsson et al., 2013*), and enhances their inhibitory innervations with anorexigenic POMC neurons (*Newton et al., 2013*), altogether resulting in hypersensitivity to starvation and increased food consumption. Interestingly, besides HFD, other metabolic challenges, including maternal HFD, alcohol consumption, as well as aging, also disrupt normal food intake via affecting the excitability and/or innervation of NPY/AgRP neurons (*Cains et al., 2017*; *Füredi et al., 2018*; *Rivera et al., 2015*). All these interventions may contribute to the onset and progression of metabolic disorders.

Before the actual food consumption, food-seeking behavior is a critical yet largely overlooked behavioral component for the localization and occupation of desirable food sources. Food-seeking behavior has been characterized in rodent models, primarily by the elevation of locomotor activity and increased food approach of starved animals (*Davidson, 2009*; *Mistlberger, 2011*). It has been reported that NPY/AgRP neurons also play a role in food-seeking behavior (*Aponte et al., 2011*). However, to ensure adequate food intake, food seeking and food consumption are temporally and spatially separated and even reciprocally inhibited (*Betley et al., 2015*; *Chen et al., 2015*). It remains largely unclear how the neural circuitry of food seeking and food consumption segregated and independently regulated in rodent models. Furthermore, it remains unknown whether HFD also affects food seeking, and if so whether its effects on both food seeking and food consumption share common mechanisms or not. To fully understand the intervention of energy homeostasis by sustained metabolic stress, we need to dissect the neural circuitry underlying food seeking and examine whether and how it is affected by HFD.

Fruit flies *Drosophila melanogaster* share fundamental analogy to vertebrate counterparts on the regulation of energy homeostasis and organismal metabolism despite that they diverged several hundred million years ago (*Pandey and Nichols, 2011*; *Rajan and Perrimon, 2013*; *Reiter et al., 2001*). Therefore, it offers a good model to characterize food-seeking behavior in depth and provides insight into the regulation of energy intake and the pathogenesis of metabolic disorders in more complex organisms such as rodents and human.

Our previous work showed that fruit flies exhibited robust starvation-induced hyperactivity that was directed towards the localization and acquisition of food sources, therefore resembling an important aspect of food-seeking behavior upon starvation (*Yang et al., 2015*). We also identified a small subset of OA neurons in the fly brain that specifically regulated starvation-induced hyperactivity (*Yu et al., 2016*). Analogous to mammalian systems, a number of neural and hormonal cues are involved in the systemic control of nutrient metabolism and food intake in fruit flies. Among them, Neuropeptide F (NPF), short NPF (sNPF), Leucokinin, and Allatostatin A (AstA), have been shown to regulate food consumption, all of which have known mammalian homologs that regulate food intake (*Pool and Scott, 2014*). In particular, starvation-induced hyperactivity is regulated by two classes of neuroendocrine cells (*Yu et al., 2016*). One is functionally analogous to pancreatic α cells and

produce AKH upon starvation, whereas the other produces *Drosophila* insulin-like peptides (DILPs), resembling the function of pancreatic β cells. These two classes of *Drosophila* hormones exert antagonistic functions on starvation-induced hyperactivity via the same group of OA neurons in the fly brain (*Yu et al., 2016*).

Based on these findings, we therefore sought to examine whether HFD disrupted the regulation of starvation-induced hyperactivity in fruit flies and aimed to investigate the underlying mechanism. In this present study, we found that HFD-fed flies became significantly more sensitive to starvation and exhibited starvation-induced hyperactivity earlier and stronger than flies fed with normal diet (ND). Meanwhile, HFD did not alter flies' food consumption, suggesting that starvation-induced hyperactivity and food consumption are independently affected by HFD. Several days of HFD treatment did not alter the production of important hormonal cues like AKH and DILPs, but rather increased the sensitivity of the OA neurons that regulated starvation-induced hyperactivity to the hunger hormone AKH. In these OA neurons, constitutive autophagy maintained the homeostasis of AKHR protein, which determined their sensitivity to AKH and hence starvation. HFD feeding suppressed neuronal autophagy via AMPK-TOR signaling and in turn increased the level of AKHR in these OA neurons. Consistently, eliminating autophagy in these neurons mimicked the effect of HFD on starvation-induced hyperactivity whereas promoting autophagy inhibited the induction of hyperactivity by starvation. Furthermore, we also showed that a specific lipoprotein LTP and its cognate receptor LpR1 likely mediated the effect of HFD on the neuronal autophagy of OA neurons and hence its effect on starvation-induced hyperactivity. Taken together, we uncovered a novel mechanism that linked HFD, AMPK-TOR signaling, neuronal autophagy, and starvation-induced hyperactivity, shedding crucial light on the reshaping of neural circuitry under metabolic stress and the progression of metabolic diseases.

## Results

### HFD specifically enhanced starvation-induced hyperactivity

We previously reported that starvation induced hyperactivity in fruit flies (*Drosophila melanogaster*). In this present study, we first confirmed that starvation-induced hyperactivity was a behavioral trait observed across different wild-type fly strains, though its magnitude might differ (*Figure 1—figure supplement 1*). In addition, the timing of the onset of starvation-induced hyperactivity might differ across different wild-type strains, too. In some strains the response started in Day 1 after starvation while some started in Day 2 (*Figure 1—figure supplement 1*). These results suggest that starvation-induced hyperactivity is a common behavioral phenomenon albeit considerable variabilities.

We then asked whether HFD affected starvation-induced hyperactivity in fruit flies. Flies were fed with ND or HFD (by adding 20% coconut oil in ND) for five consecutive days before tested in a locomotion assay that measured the frequency to cross the midline of tubes in the *Drosophila* Activity Monitor System (DAMS, Trikinetics) (*Figure 1A–B*). The midline-crossing frequency assayed by DAMS offered a reliable measure of flies' locomotor activity and hence starvation-induced hyperactivity (*Yang et al., 2015*; *Yu et al., 2016*). ND-fed flies exhibited a robust increase in locomotion upon starvation in both Day 1 and Day 2 (*Figure 1A, E–F and H–I*, blue). Compared to ND-fed flies, flies fed with HFD exhibited further enhanced hyperactivity upon starvation in both days (*Figure 1B, E–F and H–I*, orange). Meanwhile, the baseline activity of fed flies was not affected by HFD feeding (*Figure 1A–B, D and G*). These results suggest that HFD-fed flies are behaviorally more sensitive to starvation. We also compared the effect of adding 5%, 10%, and 20% coconut oil in ND and found that the effect of HFD to enhance starvation-induced hyperactivity was dose-dependent (*Figure 1—figure supplement 2*).

We have previously shown that starvation-induced hyperactivity was a reliable yet indirect measure of food-seeking behavior in fruit flies (*Yang et al., 2015*; *Yu et al., 2016*).To examine whether HFD feeding indeed modulated food-seeking behavior, we conducted more detailed analysis of HFD-fed vs. ND-fed flies in a behavioral chamber in the presence of food sources (*Figure 1—figure supplement 3A*; *Tian and Wang, 2018*). We found that HFD-fed flies exhibited significantly increased walking speed and significantly decreased latency before locating and occupying the food sources (*Figure 1—figure supplement 3B–C*), whereas their total visits to food and the duration to

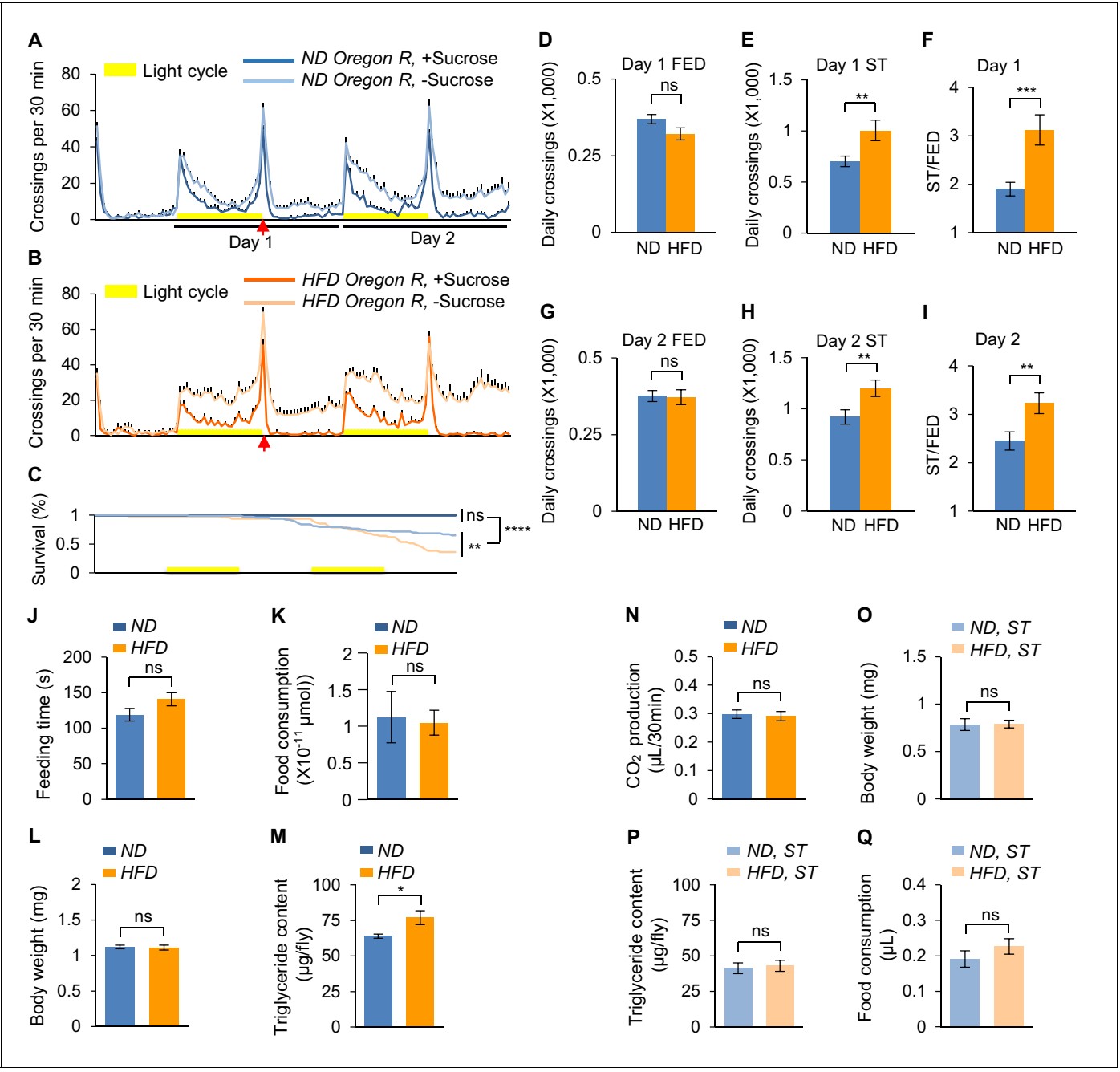

**Figure 1.** HFD promotesstarvation-induced hyperactivity in adult *Drosophila*. (**A–B**) Wild-type *Oregon-R* virgin female flies fed with normal fly food (A, ND, blue) or high-fat diet (B, HFD, orange) were assayed in the presence (dark color) and absence (light color) of 5% sucrose using DAMS-based locomotion assay (n = 46–63). Midline crossing activities of indicated flies are shown. Yellow bars represent 12 hr light-on period in this and following figures. (**C**) Survival curves of different fly groups assayed in A-B. (**D–I**) Average daily midline crossing activity of fed flies (D and G), starved flies (E and H), and starvation-induced hyperactivity (F and I) of flies assayed in A-B. Data were broken down to Day 1 (D–F) and Day 2 (G–I). In the following figures only Day one data are presented. (**J**) Total duration of feeding time during 24 hr recording in the FLIC assay (n = 38–47). (**K**) 1 hr food consumption measured by the BARCODE assay (n = 12–16 biological replicates, each containing four flies). (**L**) Average body weight of indicated flies fed ad libitum (n = 6 biological replicates, each containing three flies). (**M**) Triglyceride storage of indicated flies fed ad libitum (n = 6). (**N**) 1 hr $CO_2$ production of flies fed with ND or HFD (n = 12–14 biological replicates, each containing five flies). (**O–Q**) Average body weight (O, n = 18 biological replicates, each containing 10 flies), triglyceride storage (P, n = 6), and 100 mM sucrose consumption (Q, n = 29) of indicated flies starved for 24 hr (ST: starvation, the time point marked by arrows in A and B). ns, p>0.05; *p<0.05; ***p<0.001; ****p<0.0001. Student's t-test and one-way ANOVA followed by post hoc test with Bonferroni correction were used for pair-wise and multiple comparisons, respectively.

The online version of this article includes the following source data and figure supplement(s) for figure 1:

*Figure 1 continued on next page*

*Figure 1 continued*

**Source data 1.** Raw data of the behavioral experiments shown in *Figure 1* and *Figure 1—figure supplements 1–5*.
**Figure supplement 1.** Starvation-induced hyperactivity across different wild-type fly strains.
**Figure supplement 2.** HFD promotes starvation-induced hyperactivity in a dose-dependent manner.
**Figure supplement 3.** HFD feeding enhanced starvation-induced food seeking.
**Figure supplement 4.** Dietary saturated fat promotes starvation-induced hyperactivity.
**Figure supplement 5.** Dietary sugar does not affect starvation-induced hyperactivity.

stay on food remained unchanged (*Figure 1—figure supplement 3D–E*). These data further confirm that HFD feeding enhanced starvation-induced food seeking.

We then examined what specific aspect of HFD food induced such behavioral changes. We added three types of dietary lipid, coconut oil, soybean oil, and lard, in ND. Coconut oil and lard, which were rich in saturated fat (*Segarra et al., 2002*), significantly enhanced starvation-induced hyperactivity (*Figure 1—figure supplement 4*). In contrast, soybean oil, which was mostly unsaturated fat (*Henkel et al., 2018*), did not affect starvation-induced hyperactivity (*Figure 1—figure supplement 4*). High-sugar diet (HSD) did not enhance starvation-induced hyperactivity, either (*Figure 1—figure supplement 5*). These results suggest that excess intake of dietary saturated fat may be the cause of the observed behavioral change.

We also asked whether HFD feeding modulated other starvation-induced behavioral changes, especially food consumption. In contrast, ND-fed and HFD-fed flies exhibited comparable feeding activity in the FLIC assay (*Ro et al., 2014*) and the BARCODE assay (*Park et al., 2018*; *Figure 1J and K*). These results suggest that feeding behavior of HFD-fed flies remains unaffected, which is consistent with our previous findings that food seeking and food consumption are independently regulated (*Yang et al., 2015*; *Yu et al., 2016*).

We then sought to understand how HFD feeding enhanced starvation-induced hyperactivity. One possible explanation was that HFD might decrease flies' energy storage and/or increase flies' energy expenditure, reducing their resistence to starvation (*Figure 1C*). However, flies' body weight was not affected by HFD, and their lipid storage was actually significantly elevated by HFD (*Figure 1L and M*), a likely consequence of the uptake of more nutritious food, HFD. The energy expenditure of flies was not affected by HFD. Evidently, when fed ad libitum, the $CO_2$ production of HFD-fed and ND-fed flies was comparable (*Figure 1N*). Therefore, it is likely that HFD-fed flies are behaviorally more sensitive to starvation despite their elevated energy storage.

Consistently, upon 24 hr starvation, the body weight and lipid storage of both ND-fed and HFD-fed flies decreased (*Figure 1O and P*), whereas the food consumption of these two groups of flies increased in the MAFE assay, in comparable manners (*Figure 1Q*, note that flies exhibited no food ingestion in the MAFE assay when fed ad libitum) (*Qi et al., 2015*). Taken together, these results suggest that five days of HFD feeding specifically enhances starvation-induced hyperactivity without altering flies' food intake and energy expenditure.

## HFD increased the sensitivity of hunger-sensing, AKHR$^+$ neurons

We showed in a previous study that starvation-induced hyperactivity of adult flies was driven by a specific group of OA neurons in the fly brain (*Yu et al., 2016*). As a hunger sensor, the activity of these OA$^+$ neurons is regulated in opposite directions, by two groups of functionally antagonizing hormones, the hunger-induced hormone AKH and the satiety hormones DILPs. Since HFD enhanced starvation-induced hyperactivity despite elevated energy storage, we reasoned that HFD might exert its behavioral effect by increasing the activity of these specific OA neurons via modulating the signaling of these two groups of hormones.

We first tested whether the expression of AKH and DILP2 (the major type of DILPs expressed in the fly brain) was modulated by HFD feeding. As shown in *Figure 2A–B*, ND-fed and HFD-fed flies showed comparable AKH and DILP2 mRNA production as revealed by both RNAseq (*Figure 2A*) and quantitative RT-PCR (*Figure 2B*). Moreover, circulating AKH and DILP2 in flies' hemolymph were elevated and reduced upon starvation, respectively (*Figure 2C–D*), consistent with their roles as hunger and satiety hormones. Nevertheless, their levels remained unaffected by HFD feeding under both fed and starved conditions, as shown by the dot blot using antibodies against AKH and

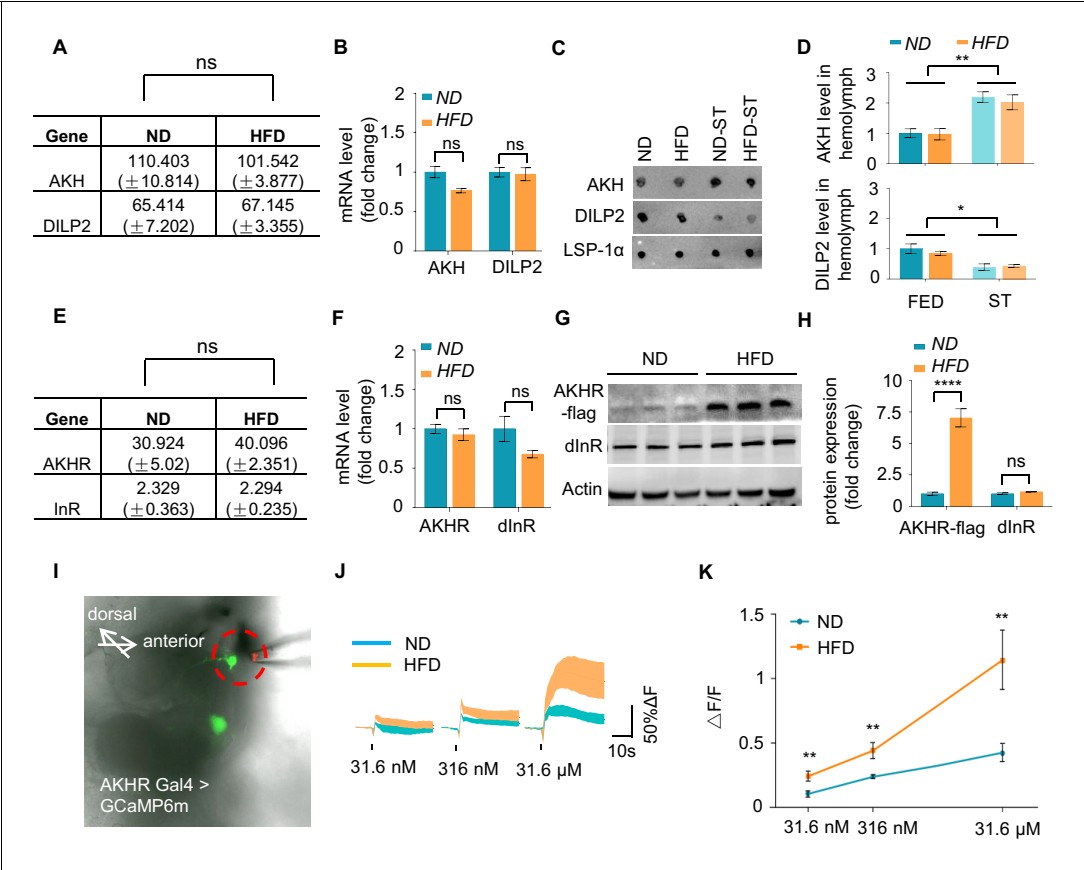

**Figure 2.** HFD increases neuronal AKHR protein. (A–B) AKH and DILP2 mRNA levels of wild-type virgin female flies fed with ND or HFD. The fly heads and associated tissues were collected and subjected to RNAseq (A) or quantitative RT-PCR (B) (n = 3 biological replicates, each containing 25 flies). (C–D) AKH and DILP2 protein levels in the hemolymph of wild-type virgin female flies fed with ND or HFD and assayed under both fed and starved conditions (n = 3 biological replicates, each containing 30 flies). Hemolymph samples were collected from the flies and AKH and DILP2 protein levels were analyzed by dot blot with AKH and DILP2 antibodies (C), and the quantification was shown in (D). (E–F) AKHR and InR mRNA levels were analyzed similar to A-B (n = 3 biological replicates, each containing 25 flies). (G–H) AKHR and dInR protein levels of fly heads were analyzed by western blot (n = 3 biological replicates, each containing 25 flies). Note that AKHR-flag knock-in flies were generated by inserting FLAG sequence into the C terminal of AKHR gene via CRISPR/cas9 mediated gene editing. Antibodies against FLAG and dInR were used in the western blot. (I) Schematic diagram of the ex vivo calcium imaging. Green signals indicate AKHR[+] neurons expressing GCaMP6 (*AKHR:BD/+; nSyb:AD/UAS-GCaMP6m*). Dotted red circle indicates the position of the pipette delivering synthetic AKH. (J–K) Representative traces (J) and quantification (K) of peak calcium transients of AKHR[+] neurons upon AKH administration (n = 7–10). ns, p>0.05; *p<0.05; ***p<0.001. Student's t-test and one-way ANOVA followed by post hoc test with Bonferroni correction were used for pair-wise and multiple comparisons, respectively.

The online version of this article includes the following figure supplement(s) for figure 2:

**Figure supplement 1.** HFD increases neuronal AKHR protein accumulation in both fed and starved conditions.

DILP2 proteins (*Figure 2C–D*). Therefore, the production of these hormones that regulated starvation-induced hyperactivity was not regulated by HFD feeding.

We then asked whether the cognate receptors of AKH and DILP2, named AKHR and dInR (*Drosophila* insulin-like receptor), were modulated by HFD feeding. While AKHR mRNA levels remained unchanged after HFD feeding, as revealed by RNAseq and quantitative RT-PCR (*Figure 2E–F*), AKHR protein was significantly elevated in the head tissue of HFD-fed flies vs. that of ND-fed flies, as demonstrated by western blot (*Figure 2G–H*). Similar results were also observed in starved flies after HFD feeding (*Figure 2—figure supplement 1*). The production of dInR was not affected by HFD at both the transcriptional and translational levels (*Figure 2E–H*). These data suggest that HFD exerts a robust effect on AKHR protein accumulation without altering its mRNA production in the fly head.

We have previously shown that AKHR was expressed in two clusters of OA neurons in the subesophageal zone (SEZ) region of the fly brain (*Yu et al., 2016*). These neurons were responsive to the hunger hormone AKH and exerted a robust effect in inducing starvation-induced hyperactivity (*Yu et al., 2016*). Since AKHR protein levels were upregulated by HFD feeding, we reasoned that these AKHR$^+$ neurons might be more sensitive to AKH upon HFD feeding. Indeed, by using a calcium imaging setup with ex vivo brain preparations, we found that these AKHR$^+$ neurons in HFD-fed flies exhibited more robust calcium transients when stimulated with synthetic AKH than those in ND-fed flies in a dose-dependent manner (*Figure 2I–K*). Taken together, HFD leads to increased AKHR protein accumulation in the fly brain and enhanced sensitivity of AKHR$^+$ neurons to the hunger hormone AKH. These results may account for the observations that HFD-fed flies become more sensitive to starvation and exhibit enhanced hyperactivity upon starvation (*Figure 1*).

## HFD enhanced AKHR accumulation by suppressing neuronal autophagy

We then sought to understand how HFD enhanced AKHR protein accumulation without promoting its mRNA production. One plausible hypothesis was that the degradation of AKHR via lysosome and/or proteasome was suppressed by HFD feeding. To examine this possibility, we ectopically expressed HA-tagged AKHR protein in cultured *Drosophila* S2 cells, and found that blocking lysosome-mediated protein degradation by lysosome inhibitors chloroquine (CQ) and NH$_4$Cl increased AKHR protein levels dramatically, while MG132 treatment to disturb proteasome function exerted little effect (*Figure 3A–B*). Therefore, AKHR may be mainly degraded via the lysosome pathway.

We next asked whether AKHR was degraded via autophagy, a specific lysosome-dependent protein degradation pathway (*Klionsky, 2007*). When treated with rapamycin, a robust inducer of autophagy, cultured S2 cells quickly formed characteristic autophagosomes, as indicated by the formation of dotted fluorescent GFP-ATG8 (the fly homolog of mammalian LC3) (*Figure 3C*, green). Notably, upon rapamycin treatment, HA-tagged AKHR protein was also enriched in these autophagosomes (*Figure 3C*, red). Moreover, inducing autophagy by both starvation and rapamycin resulted in significantly decreased AKHR protein levels in cultured S2 cells (*Figure 3D–E*). Therefore, the degradation of AKHR was mainly mediated by the lysosome-dependent autophagy pathway in vitro. Notably, although it was widely accepted that membrane proteins were mainly degraded by endocytosis and then lysosome (e.g. LDLR and EGFR) (*Beguinot et al., 1984*), it was also reported that certain membrane proteins could be degraded via autophagy, such as GABA and AMPA receptor subunits (*Jin et al., 2018*).

We then aimed to confirm these in vitro results and asked whether HFD induced AKHR accumulation in AKHR$^+$ neurons via suppressing its degradation by the autophagy pathway in vivo. The ratio of membrane-associated ATG8-II over soluble ATG8-I was a reliable indicator of autophagy induction (*Mizushima et al., 2010*). Indeed, HFD suppressed autophagy in the AKHR$^+$ neurons, as evident by the relative decrease of membrane-associated ATG8-II upon HFD feeding (GFP-ATG8-II/I, *Figure 3F–G*). Additionally, specifically suppressing autophagy in AKHR$^+$ neurons, by RNAi knockdown of two critical autophagy initiators ATG5 and ATG7, significantly enhanced AKHR accumulation in ND-fed flies (*Figure 3H–I*), phenocopying the effect of HFD feeding (*Figure 2*). Taken together, HFD feeding suppresses neuronal autophagy and enhances AKHR accumulation in AKHR$^+$ neurons.

## Suppression of autophagy in AKHR$^+$ neurons enhanced their sensitivity to starvation

AKHR and the activity of AKHR$^+$ neurons are crucial for starvation-induced hyperactivity (*Yu et al., 2016*). We found that eliminating AKHR gene expression (*Figure 4—figure supplement 1*) and silencing AKHR$^+$ neurons (*Figure 4—figure supplement 2*) both led to diminished effect of HFD to enhance starvation-induced hyperactivity, and that artificial activation of AKHR$^+$ neurons (*Figure 4—figure supplement 3*) resulted in increased starvation-induced hyperactivity under ND feeding. Since HFD enhanced AKHR accumulation in AKHR$^+$ neurons via suppressing autophagy, we hypothesized that blocking autophagy in AKHR$^+$ neurons in ND-fed flies would mimic the effect of HFD feeding, leading to enhanced AKH sensitivity and hence stronger starvation-induced hyperactivity.

We then directly tested this hypothesis in the calcium imaging setup using the ex vivo brain preparation. Indeed, RNAi knock-down of ATG5 and ATG7 in AKHR$^+$ neurons both led to significantly

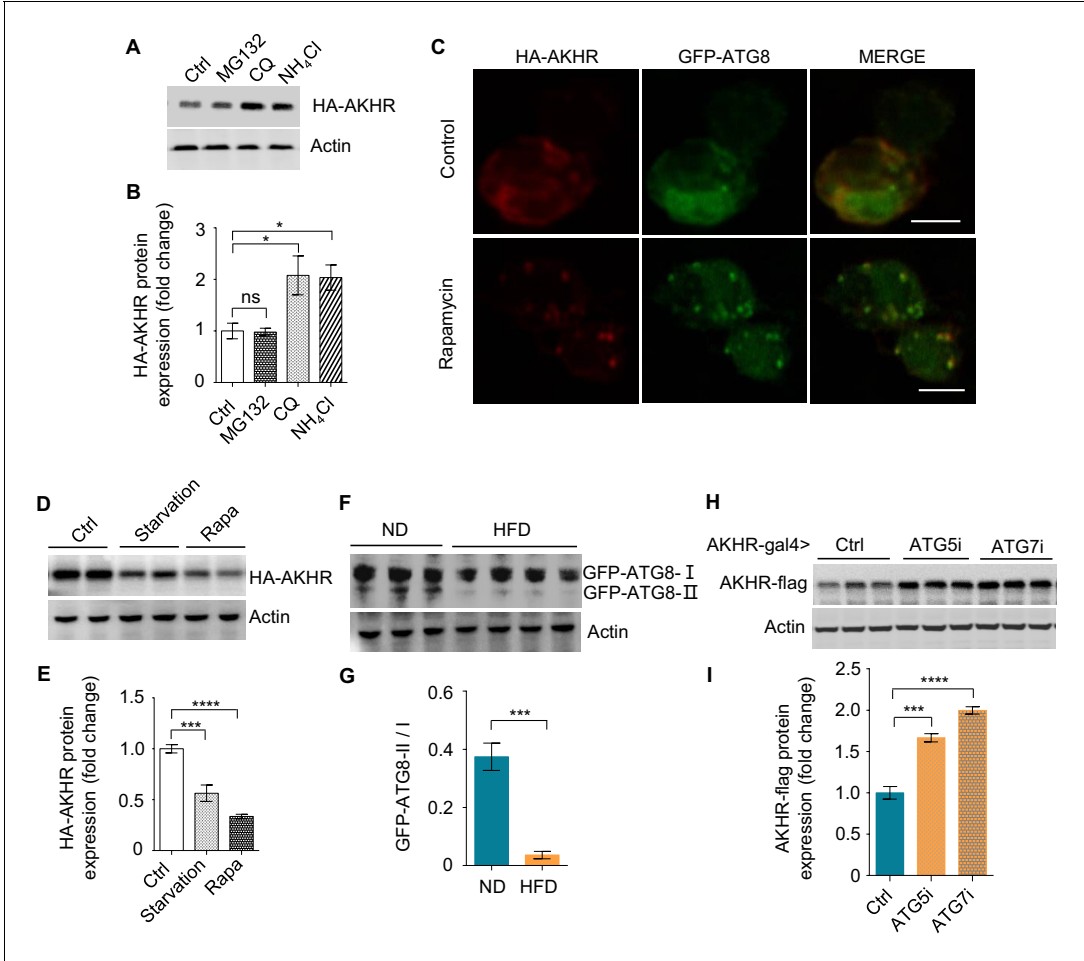

**Figure 3.** AKHR accumulation is induced by the suppression of autophagy. (**A–B**) Quantification of AKHR protein levels in cultured *Drosophila* S2 cells upon the treatment of proteasome and lysosome inhibitors (n = 3 biological replicates). Cultured cells transiently expressing AKHR-HA were treated with the indicated chemicals. The lysates were analyzed by western blot with HA antibody (**A**) and quantified in (**B**). (**C**) Formation of autophagosome in cultured S2 cells. Cultured cells transiently expressing AKHR-HA and GFP-ATG8 were treated with 200 nM rapamycin for 12 hr and stained with HA antibody. (**D–E**) Quantification of AKHR protein levels in Cultured *Drosophila* S2 cells upon starvation and the treatment of rapamycin (n = 3 biological replicates). Cultured S2 cells transiently expressing AKHR-HA were starved or treated with rapamycin, and AKHR-HA was analyzed by western blot. (**F–G**) Quantification of ATG8 activation in the head tissues of flies in vivo (n = 3–4 biological replicates, each containing 25 flies). GFP-ATG8 was specifically expressed in AKHR[+] neurons by using the GAL4-UAS system. Flies were fed with ND or HFD and their head tissues were collected for western blot with GFP antibody. The ratio between ATG8-II and ATG8-I bands indicates the level of autophagy. (**H–I**) Quantification of AKHR protein levels upon suppressing neuronal autophagy in AKHR[+] neurons in vivo (n = 3 biological replicates,, each containing 25 flies). ATG5 and ATG7 were knocked down by the expression of RNAi constructs in AKHR[+] neurons. AKHR-flag expression was assayed by western blot. ns, p>0.05; *p<0.05; **p<0.01; ****p<0.0001. Student's t-test and one-way ANOVA followed by post hoc test with Bonferroni correction were used for pair-wise and multiple comparisons, respectively.

enhanced calcium transients upon AKH administration in ND-fed flies (*Figure 4A–B*), phenocopying the effect of HFD feeding.

Consistent with these results, blocking autophagy in AKHR[+] neurons via RNAi knock-down of ATG5 significantly enhanced starvation-induced hyperactivity in ND-fed flies without altering their baseline activity (*Figure 4C–F*), again phenocopying the effect of HFD feeding. ATG7 knock-down exerted similar behavioral results (*Figure 4—figure supplement 4*). In both conditions, HFD feeding did not further enhance starvation-induced hyperactivity (*Figure 4F* and *Figure 4—figure supplement 4*), likely due to the suppression of autophagy by ATG5/7 knockdown. Taken together, suppressing neuronal autophagy in AKHR[+] neurons resembles the effect of HFD feeding, enhancing the sensitivity of AKHR[+] neurons as well as starvation-induced hyperactivity.

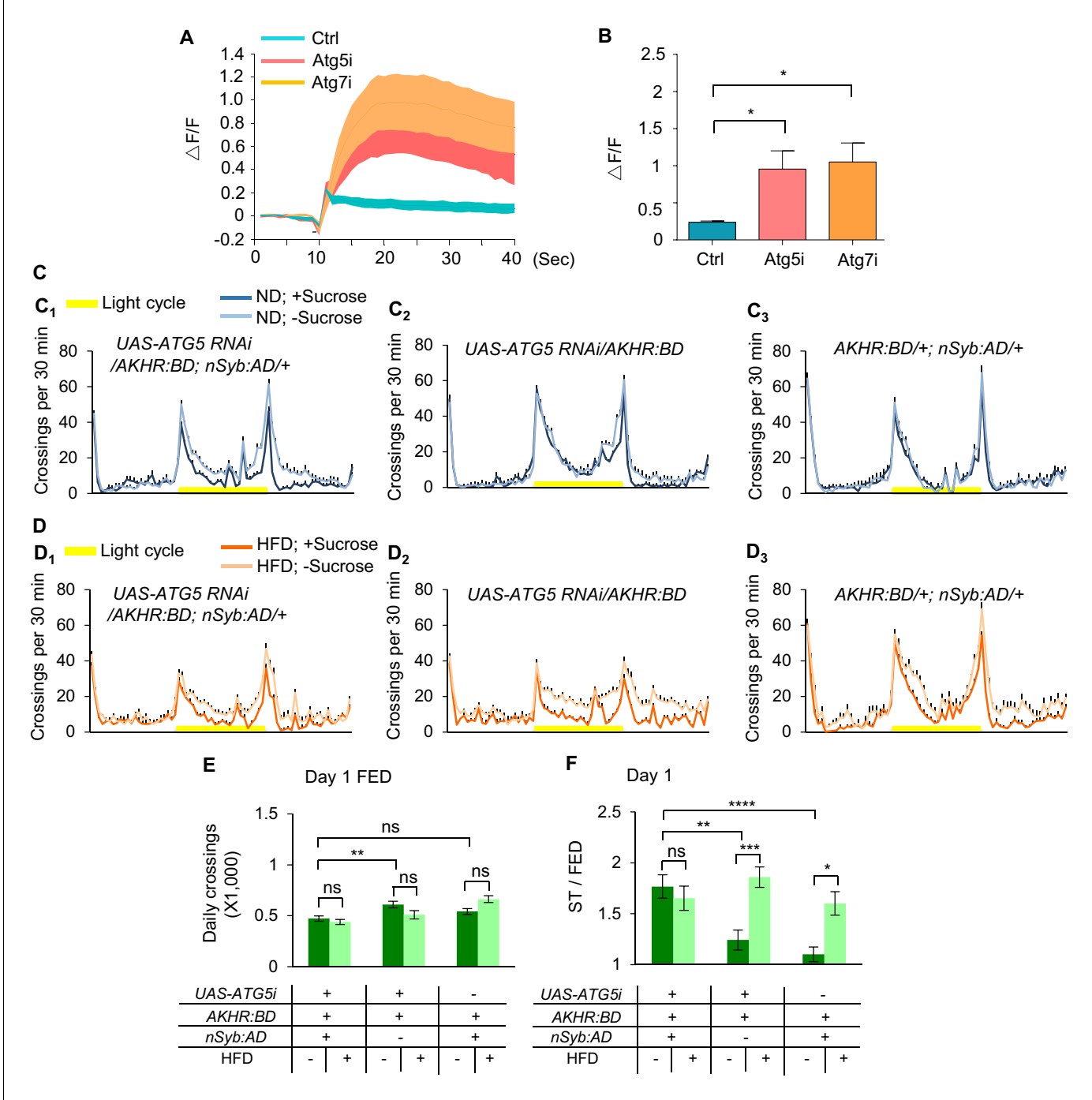

**Figure 4.** Inhibition of autophagy increases starvation-induced hyperactivity. (**A–B**) Representative traces (**A**) and quantification (**B**) of peak calcium responses of AKHR[+] neurons to 316 nM AKH upon suppressing neuronal autophagy in AKHR[+] neurons (n = 7–10). (**C–D**) Midline crossing activity of indicated genotypes and diet treatments assayed in the presence or absence of 5% sucrose (n = 43–62). (**E**) Average daily midline crossing activity of fed flies assayed in C-D. (**F**) Starvation-induced hyperactivity of flies assayed in C-D. ns, p>0.05; *p<0.05; **p<0.01; ***p<0.001; ****p<0.0001. One-way and two-wayANOVA followed by post hoc test with Bonferroni correction were used for multiple comparisons when applicable.
The online version of this article includes the following source data and figure supplement(s) for figure 4:

**Source data 1.** Raw data of the behavioral experiments shown in *Figure 4* and *Figure 4—figure supplements 1–4*.
**Figure supplement 1.** *AKHR* gene is required for the enhancement of starvation-induced hyperactivity by HFD.
**Figure supplement 2.** Silencing AKHR[+] neurons diminishes the effect of HFD feeding to enhance starvation-induced hyperactivity.
**Figure supplement 3.** Activation AKHR[+] neurons enhances starvation-induced hyperactivity under ND feeding.
**Figure supplement 4.** Knocking down ATG7 in AKHR[+] neurons promotes starvation-induced hyperactivity.

## HFD suppressed autophagy via activating AMPK-TOR signaling

To further examine the cellular mechanism underlying HFD-induced suppression of neuronal autophagy, we carried out RNAseq analysis of fly head tissues harvested from ND-fed and HFD-fed flies (*Figure 5—source data 1*; *Figure 5—figure supplements 1*). As shown in *Figure 5—figure supplement 2*, we found that a collection of genes related to anabolic activities were upregulated while a collection of catabolic genes were downregulated in HFD-fed flies. These results hinted a possibility that TOR (target of rapamycin) signaling, the key regulator of anabolism, was activated by HFD. Meanwhile, TOR is known to be a negative regulator of autophagy in both yeasts and animals (*Wullschleger et al., 2006*). Therefore, TOR signaling may mediate the effect of HFD to suppress autophagy.

To further confirm this hypothesis, we examined p70S6K phosphorylation in vivo, which was a reliable indicator of TOR activation (*Wullschleger et al., 2006*). Indeed, the head tissues of HFD-fed flies exhibited higher levels of p70S6K phosphorylation (*Figure 5A–B*), suggesting that HFD activated TOR signaling. TOR signaling could be modulated by two major upstream signals, negatively by AMPK (AMP-activated protein kinase) signaling and positively by AKT signaling (*Wullschleger et al., 2006*). We found that upon HFD feeding, AMPK signaling (measured by the phosphorylation of AMPK) was suppressed, whereas AKT signaling (measured by the phosphorylation of AKT) remained unaffected (*Figure 5C–D*), suggesting that HFD modulates TOR signaling via the suppression of AMPK signaling.

Since AMPK-TOR signaling was known to modulate autophagy and the degradation of specific proteins, it was therefore possible that HFD suppressed neuronal autophagy and hence enhanced AKHR accumulation via modulating AMPK-TOR signaling. If this was the case, manipulating AMPK-TOR signaling would exert a robust effect on AKHR accumulation just as HFD feeding. Indeed, pharmacological activation of TOR signaling by MHY and suppression of AMPK by dorsomorphin (dorso) both resulted in AKHR accumulation in cultured S2 cells, whereas suppression of TOR signaling by rapamycin and activation of AMPK by AICAR exerted the opposite effect (*Figure 5E–F*). Consistent with these in vitro results, dorso and rapamycin feeding led to increased and decreased AKH sensitivity in AKHR$^+$ neurons, respectively (*Figure 5G–H*), further confirming that AMPK-TOR signaling plays an important role in regulating AKHR accumulation and hence the sensitivity of AKHR$^+$ neurons.

## AMPK-TOR signaling mediated the effect of HFD on starvation-induced hyperactivity

Since AMPK-TOR signaling mediated the effect of HFD on AKHR accumulation and the neuronal sensitivity of AKHR$^+$ neurons, we reasoned that manipulating AMPK-TOR signaling in vivo would affect the onset of starvation-induced hyperactivity.

Indeed, suppressing TOR signaling in AKHR$^+$ neurons by ectopic expression of dominant negative TOR$^{TED}$ (*Hennig and Neufeld, 2002*) resulted in significantly decreased starvation-induced hyperactivity in ND-fed flies, and these flies did not respond to HFD (*Figure 6A–D*). These data suggest that TOR signaling is required for starvation-induced hyperactivity as well as the effect of HFD to further enhance this behavior.

To further confirm these results, we introduced additional pharmacological and genetic manipulations. Like TOR$^{TED}$ expression in AKHR$^+$ neurons, rapamycin feeding suppressed starvation-induced hyperactivity in ND-fed flies (*Figure 6—figure supplement 1*). Notably, unlike TOR$^{TED}$ expression AKHR$^+$ neurons, rapamycin feeding enhanced the baseline locomotion of fed flies (*Figure 6—figure supplement 1C*), suggesting that global suppression of TOR signaling may have a locomotor effect.

Besides, we genetically activated TOR signaling in AKHR$^+$ neurons, by RNAi knock-down of TSC1, a suppressor of TOR signaling (*Wullschleger et al., 2006*) and by the ectopic expression of a dominant negative form of AMPK (AMPK$^{DN}$) (*Wullschleger et al., 2006*). Both manipulations resulted in upregulated starvation-induced hyperactivity in ND-fed flies (*Figure 6—figure supplements 2 and 3*), phenocopying the effect of HFD feeding (*Figure 1*). Collectively, these data further confirm that HFD enhances starvation-induced hyperactivity via modulating AMPK-TOR signaling in AKHR$^+$ neurons.

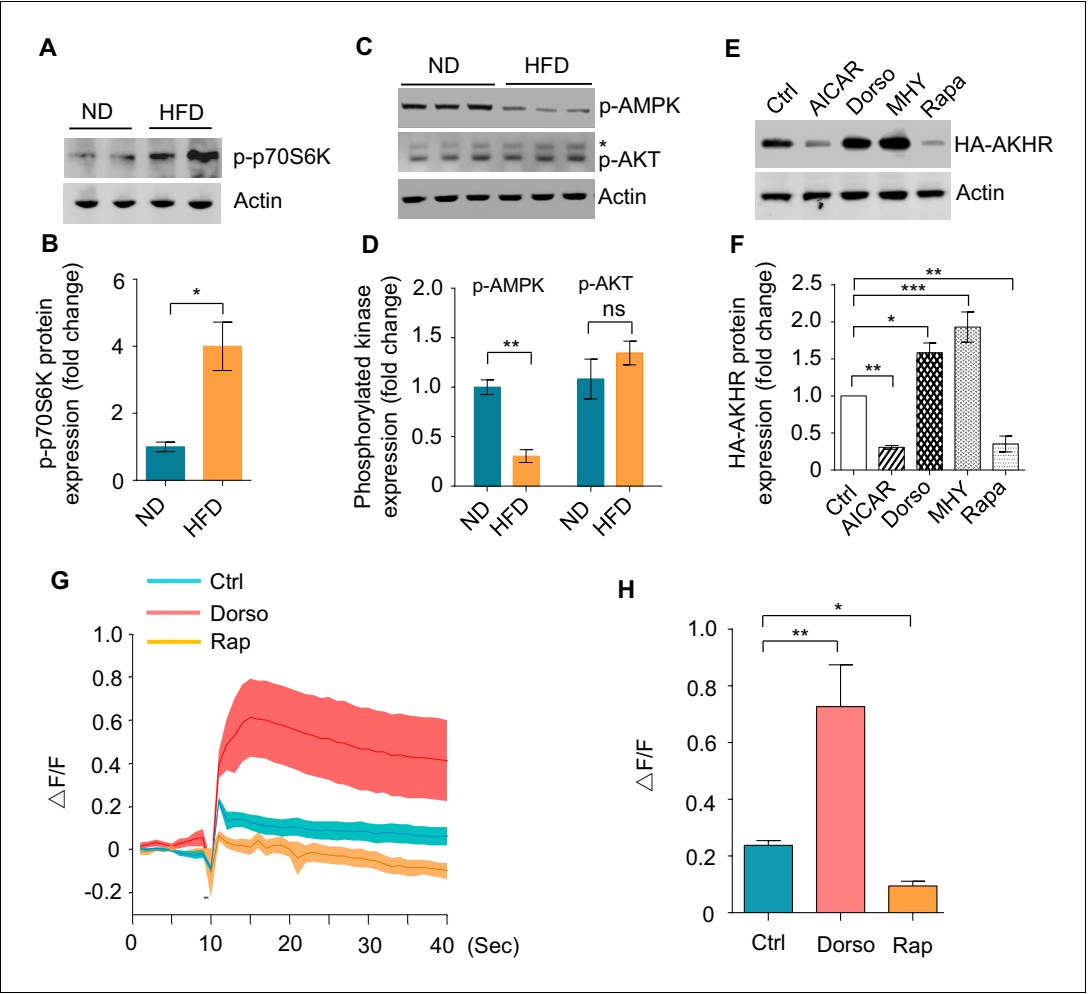

**Figure 5.** HFD activates TOR signaling. (**A–D**) p70s6k/AKT/AMPK phosphorylation under ND vs. HFD feeding conditions. The head tissues of wild-type flies fed with ND or HFD were harvested and subjected to western blot with phosphorylated p70s6k antibody (**A–B**) and p-AKT/p-AMPK antibodies (**C–D**) (n = 3 biological replicates, each containing 25 flies). *Nonspecific bands. (**E–F**) AKHR accumulation upon pharmacological manipulation of AMPK-TOR signaling in cultured *Drosophila* S2 cells (n = 3). Cultured S2 cells transiently expressing AKHR-HA were treated with the indicated chemicals and then analyzed by western blot with HA antibody. (**G–H**) Representative traces (**G**) and quantification (**H**) of peak calcium responses of AKHR[+] neurons to 316 nM AKH, with or without the indicated drug treatment (n = 7–10). ns, p>0.05; *p<0.05; **p<0.01; ***p<0.001. Student's t-test and one-way ANOVA followed by post hoc test with Bonferroni correction were used for pair-wise and multiple comparisons, respectively.

The online version of this article includes the following source data and figure supplement(s) for figure 5:

**Source data 1.** Top differentially expressed genes upon HFD feeding.
**Figure supplement 1.** KEGG pathway analysis of differentially expressed genes upon HFD feeding.
**Figure supplement 2.** TOR downstream genes are modulated by HFD feeding.

## A lipoprotein LTP and its cognate receptor LpR1 linked HFD feeding to enhanced starvation-induced hyperactivity

We next sought to understand how HFD feeding exerted a robust effect on the autophagy pathway of a small group of brain neurons expressing AKHR. Since dietary lipids needed to enter the circulating system and different fly organs via specific carriers named lipoproteins, we asked whether certain lipoprotein(s) played a role in mediating the behavioral effect of HFD feeding.

Three lipophorins, the main *Drosophila* lipoprotein, circulate in the hemolymph and transport lipids to different fly organs (*Rodríguez-Vázquez et al., 2015*). We performed proteomic analysis of

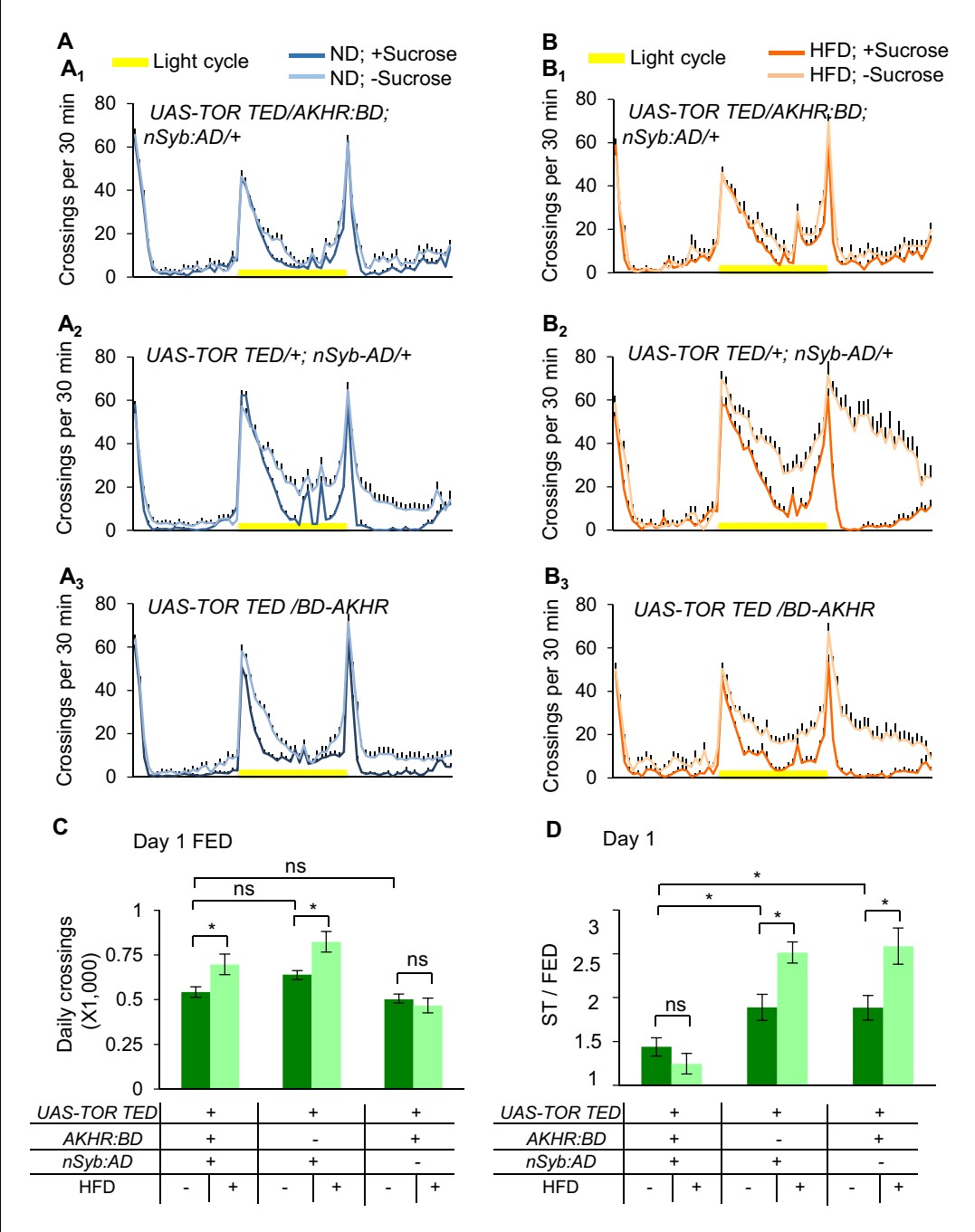

**Figure 6.** AMPK-TOR signaling modulates starvation-induced hyperactivity. (**A–B**) Midline crossing activity of indicated genotypes and diet treatments assayed in the presence or absence of 5% sucrose (n = 35–58). (**C**) Average daily midline crossing activity of fed flies assayed in A-B. (**D**) Starvation-induced hyperactivity of flies assayed in A-B. ns, p>0.05; *p<0.05; **p<0.01; ***p<0.001. One-way and two-way ANOVA followed by post hoc test with Bonferroni correction were used for multiple comparisons when applicable.

The online version of this article includes the following source data and figure supplement(s) for figure 6:

**Source data 1.** Raw data of the behavioral experiments shown in *Figure 6* and *Figure 6—figure supplements 1–3*.
**Figure supplement 1.** Rapamycin feeding suppresses starvation-induced hyperactivity.
**Figure supplement 2.** Knocking down TSC1 in AKHR+ neurons promotes starvation-induced hyperactivity.
**Figure supplement 3.** Suppressing AMPK signaling in AKHR+ neurons promotesstarvation-induced hyperactivity.

circulating proteins in flies' hemolymph by mass spectrometry (*Figure 7—figure supplement 1*; *Figure 7—source data 2*) and found that only one of the three lipoproteins, named Lipid Transfer Particle (LTP), but not the other two [Lipophorin (LPP) and Microsomal Triglyceride Transfer Protein (MTP)], was significantly upregulated in HFD-fed flies vs. ND-fed flies (*Figure 7A–B*). Meanwhile, both triglycerides and cholesterol were enriched in the hemolymph of HFD-fed flies (*Figure 7C–D*). Therefore, it was possible that LTP was the major carrier for excess circulating lipids after HFD feeding.

Recruitment of lipoproteins to the cell membrane by its receptors is a key event that initiates the transfer of neutral lipids to cells (*Rodríguez-Vázquez et al., 2015*). To understand the role of excess circulating lipids in suppressing autophagy of AKHR$^+$ neurons, it was necessary to identify lipoprotein receptor expressed in these cells. To this aim, we carried out single cell RNAseq analysis for individual AKHR$^+$ neurons harvested from the fly brains in situ. Among seven identified lipophorin receptors in *Drosophila*, lipophorin receptor 1 (LpR1) was most abundantly expressed in AKHR$^+$ neurons (*Figure 7E*).

We thus examined the potential role of LpR1 in mediating the effect of HFD feeding on AKHR accumulation and starvation-induced hyperactivity. RNAi knock-down of LpR1 in AKHR$^+$ neurons eliminated the effect of HFD on AKHR accumulation (*Figure 7F–G*). Consistently, these flies did not respond to HFD feeding with an enhancement of starvation-induced hyperactivity (*Figure 7H*). But it is worth noting that these flies exhibited a slightly higher (yet insignificant) starvation-induced hyperactivity when fed with ND, too (*Figure 7H$_5$*). Meanwhile, these flies also exhibited an increase in their baseline locomotion under fed conditions (*Figure 7H$_4$*). These behavioral confounds made it difficult to distinguish whether LpR1 knock-down resulted in higher starvation-induced hyperactivty, or blunted response to HFD, or both. Nevertheless, it was clear that LpR1 in AKHR$^+$ neurons was required for HFD to modulate starvation-induced hyperactivty.

Therefore, a specific lipoprotein LTP and its cognate receptor LpR1 may mediate the effect of HFD on the onset of starvation-induced hyperactivity. Detailed mechanistic studies of LTP-LpR1 signaling are required to fully elucidate their functions in regulating neuronal autophagy and AKHR protein accumulation upon HFD feeding.

## Diet-independent hyperlipidemia also enhanced starvation-induced hyperactivity

HFD is known to be a robust reward signal and can induce food craving (*de Macedo et al., 2016*). We therefore asked whether enhanced starvation-induced hyperactivity by HFD was solely mediated by excess lipid uptake, or HFD feeding also exerted other non-metabolic effects. By knocking down LpR1 in the fat body, the major organ for lipid storage and metabolism (*Zheng et al., 2016*), we generated diet-independent hyperlipidemia in ND-fed flies (*Figure 8A–B*). Notably, similar effect was also observed in mouse models (*Wang et al., 2016*).

We then asked whether diet-independent hyperlipidemia had similar effects on AKHR accumulation and starvation-induced hyperactivity as HFD feeding. Compared with the control group, the phosphorylation of p70S6K in the diet-independent hyperlipidemia group was significantly up-regulated (*Figure 8C–D*). These results showed that like HFD feeding, diet-independent hyperlipidemia also activated TOR signaling. In addition, diet-independent hyperlipidemia led to upregulated AKHR accumulation (*Figure 8E–F*), mimicking the effect of HFD feeding. Consistent with these results, these ND-fed, hyperlipidemic flies showed significantly stronger starvation-induced hyperactivity than the two control groups (*Figure 8G*), which was comparable to that of HFD-fed flies (*Figure 8G*). Taken together, diet-independent hyperlipidemia phenocopied the effect of HFD feeding, suggesting that HFD feeding likely induces hyperlipidemia in fruit flies, which subsequently results in the cellular and circuitry changes underlying enhanced starvation-induced hyperactivity.

## Discussion

There is accumulating evidence that notes the effect of HFD on food consumption from insects to human, which results in obesity and obesity-associated metabolic diseases. But the effect of HFD on another critical food intake related behavior, food seeking, remains largely uncharacterized. Conceptually, food-seeking behavior in the fruit fly is composed of two behavioral components, increased sensitivity to food cues, and enhanced exploratory locomotion, which altogether facilitates the

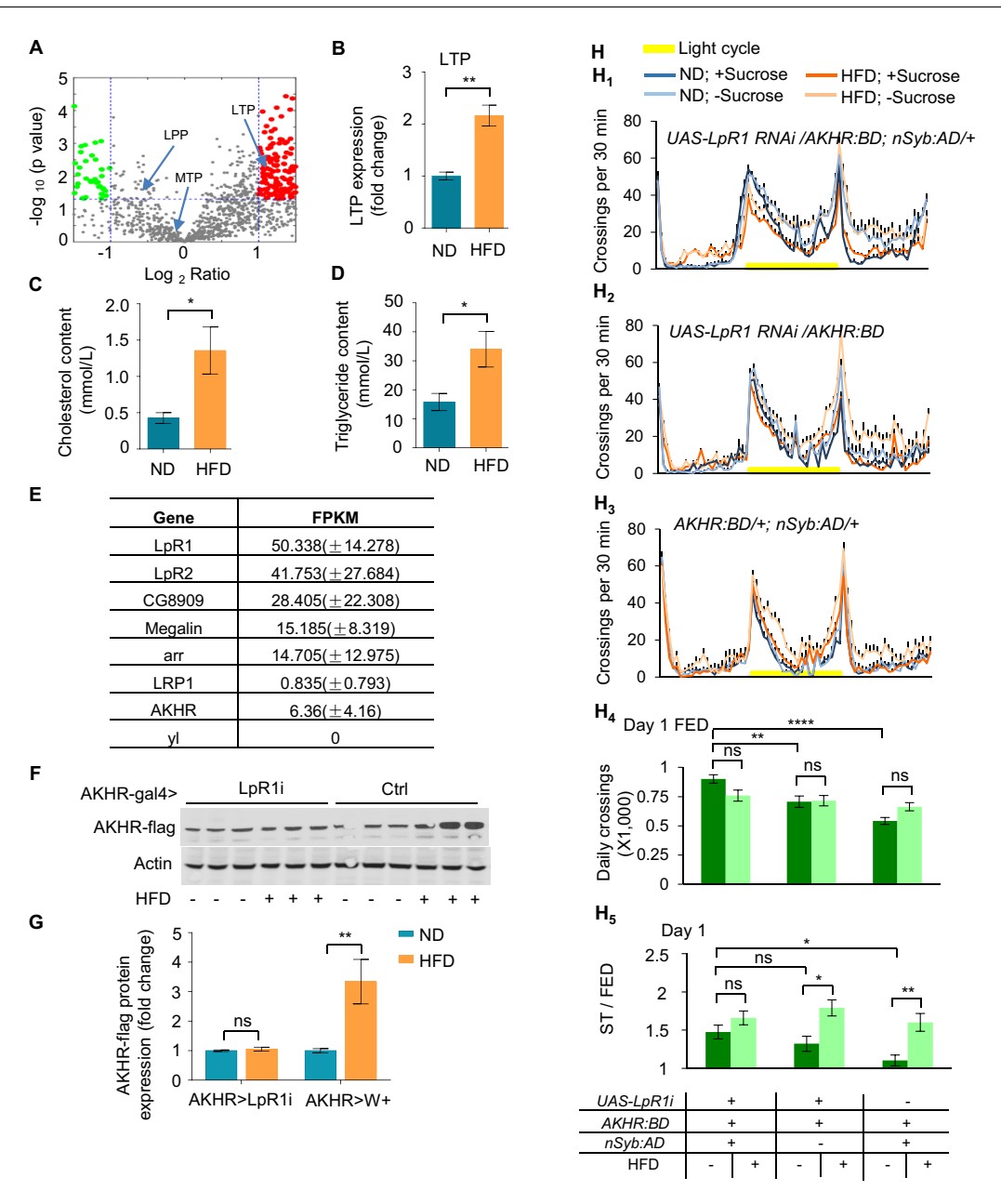

**Figure 7.** LpR1 is required for HFD-strengthened hyperactivity under starvation. (**A–B**) LTP protein levels in the hemolymph. Hemolymph was collected from flies fed with ND or HFD and subjected to LC-MS/MS analysis. Volcano plot (**A**) shows peptides with differential enrichment (under ND vs. HFD feeding conditions). The horizontal line indicates the significance threshold (p=0.01) and the vertical lines indicate two-fold change. Green and red dots represent down-regulated and up-regulated peptides, respectively. Arrows indicate three lipoproteins (LTP, LPP and MTP). The level of LTP was shown in B (n = 3 biological replicates, each containing ~300 flies). (**C–D**) The cholesterol level (**C**) and triglyceride level (**D**) in the hemolymph harvested from ND vs. HFD fed flies (n = 4–6 biological replicates, each containing 60 flies). (**E**) Single cell RNAseq shows the gene expression level of candidate lipoprotein receptors in AKHR[+] neurons (n = 5 cells). (**F–G**) AKHR protein levels from the head tissues of the indicated genotypes fed with ND or HFD (n = 3 biological replicates, each containing 25 flies). (**H1–H3**) Midline crossing activity of indicated genotypes and diet treatments assayed in the presence or absence of 5% sucrose (n = 43–57). (**H4**) Average daily midline crossing activity of fed flies assayed in H1-H3. (**H5**) Starvation-induced hyperactivity of flies assayed in H1-H3. ns, p>0.05; *p<0.05; **p<0.01. Student's t-test, one-way and two-way ANOVA followed by post hoc test with Bonferroni correction were used for multiple comparisons when applicable.

The online version of this article includes the following source data and figure supplement(s) for figure 7:

**Source data 1.** Raw data of the behavioral experiments and mass spec experiments shown in *Figure 7*.

**Source data 2.** Top differentially expressed peptides upon HFD feeding.

*Figure 7 continued on next page*

*Figure 7 continued*

**Figure supplement 1.** KEGG pathway analysis of differentially expressed peptides upon HFD feeding.

localization and acquisition of desirable food sources (*Root et al., 2011*; *Yu et al., 2016*). Previous work from our lab has shown that starvation promotes starvation-induced hyperactivity, the exploratory component of food-seeking behavior, via a small group of OA neurons in the fly brain. These hunger-sensing OA neurons sample the metabolic status by detecting two groups of functionally antagonistic hormones, AKH and DILPs, and promote starvation-induced hyperactivity (*Yu et al., 2016*).

In this present study, we demonstrated that this behavior was compromised by metabolic challenges. After a few days of HFD feeding, flies became behaviorally hypersensitive to starvation and as a result their starvation-induced hyperactivity was greatly enhanced, despite that their food intake and expenditure were not affected. These results suggest that HFD feeding may specifically modulate the activity of the neural circuitry underlying starvation-induced hyperactivity and offer us an opportunity to further elucidate the cellular and circuitry mechanisms underlying behavioral abnormalities upon metabolic challenges (*Figure 8H*).

As an insect counterpart of mammalian glucagon, AKH acts as a hunger signal to activate its cognate receptor AKHR expressed in the fat body and subsequently triggers lipid mobilization and energy allocation (*Bharucha et al., 2008*). In the fly brain, a small number of OA neurons also express AKHR. We and other labs have shown that these neurons are responsive to starvation and modulate various behaviors including food seeking and drinking (*Jourjine et al., 2016*; *Yu et al., 2016*). In that sense, these OA$^+$AKHR$^+$ neurons are functionally analogous to mammalian NPY/AgRP neurons in the hypothalamus, which also senses organismal metabolic states and regulates specific food intake behaviors. In this present study, we found that OA$^+$AKHR$^+$ neurons exhibited higher AKHR protein accumulation and became hypersensitive to AKH after HFD feeding. Notably, HFD feeding in mammals also increases the excitability of NPY/AgRP neurons, which contributes to the hypersensitivity to starvation and increased food consumption (*Vernia et al., 2016*). Thus, HFD may exert a conserved effect in the regulation of neuronal excitability and food intake related behaviors in both fruit flies and mammals.

Autophagy, a lysosomal degradative process that maintains cellular homeostasis, is critical for energy homeostasis. Upon cellular starvation, autophagy generates additional energy supply by breaking down macromolecules and subcellular organelles (*Klionsky, 2007*). At the organismal level, autophagy also contributes to the regulation of food intake and hence organismal energy homeostasis. For example, fasting induces autophagy in NPY/AgRP neurons via fatty acid uptake and promotes AgRP expression, which in turn enhances food intake (*Kaushik et al., 2011*). In line with these results, eliminating autophagy in NPY/AgRP neurons reduces food intake and hence body weight and fat deposits (*Kaushik et al., 2011*). Conversely, loss of autophagy in POMC neurons displays increased food intake and adiposity (*Coupé et al., 2012*). Consistently, in this present study, we showed that in fruit flies neuronal autophagy was critical for the function of OA$^+$AKHR$^+$ neurons to sense hunger and regulate starvation-induced hyperactivity.

Accumulating evidence suggests that HFD suppresses autophagy in different peripheral tissue types such as liver, skeletal muscle, and the adipose tissue (*Feng et al., 2016*; *He et al., 2012*; *Liu et al., 2009*). Similarly, HFD suppresses autophagy in the hypothalamus, whereas blocking hypothalamic autophagy, particularly in POMC neurons, exacerbates HFD induced obesity (*Coupé et al., 2012*). In this present study, we showed that HFD suppressed neuronal autophagy in OA$^+$AKHR$^+$ neurons and enhanced AKHR accumulation in these neurons. As a result, OA$^+$AKHR$^+$ neurons became hypersensitive to starvation and promoted starvation-induced hyperactivity. It will be of interest to examine whether HFD also reduces autophagy and increases the accumulation of specific membrane receptors in mammalian NPY/AgRP neurons.

We also sought to examine the cellular mechanism that linked HFD feeding to the reduction of autophagy. We found that HFD feeding activated TOR signaling. TOR, a highly conserved serine-threonine kinase, controls numerous anabolic cellular processes (*Proud, 2007*). We found in this present study that TOR signaling was tightly associated with the activity of AKHR$^+$ neurons and the behavioral responses upon HFD feeding. Genetic enhancement of TOR activity in AKHR$^+$ neurons

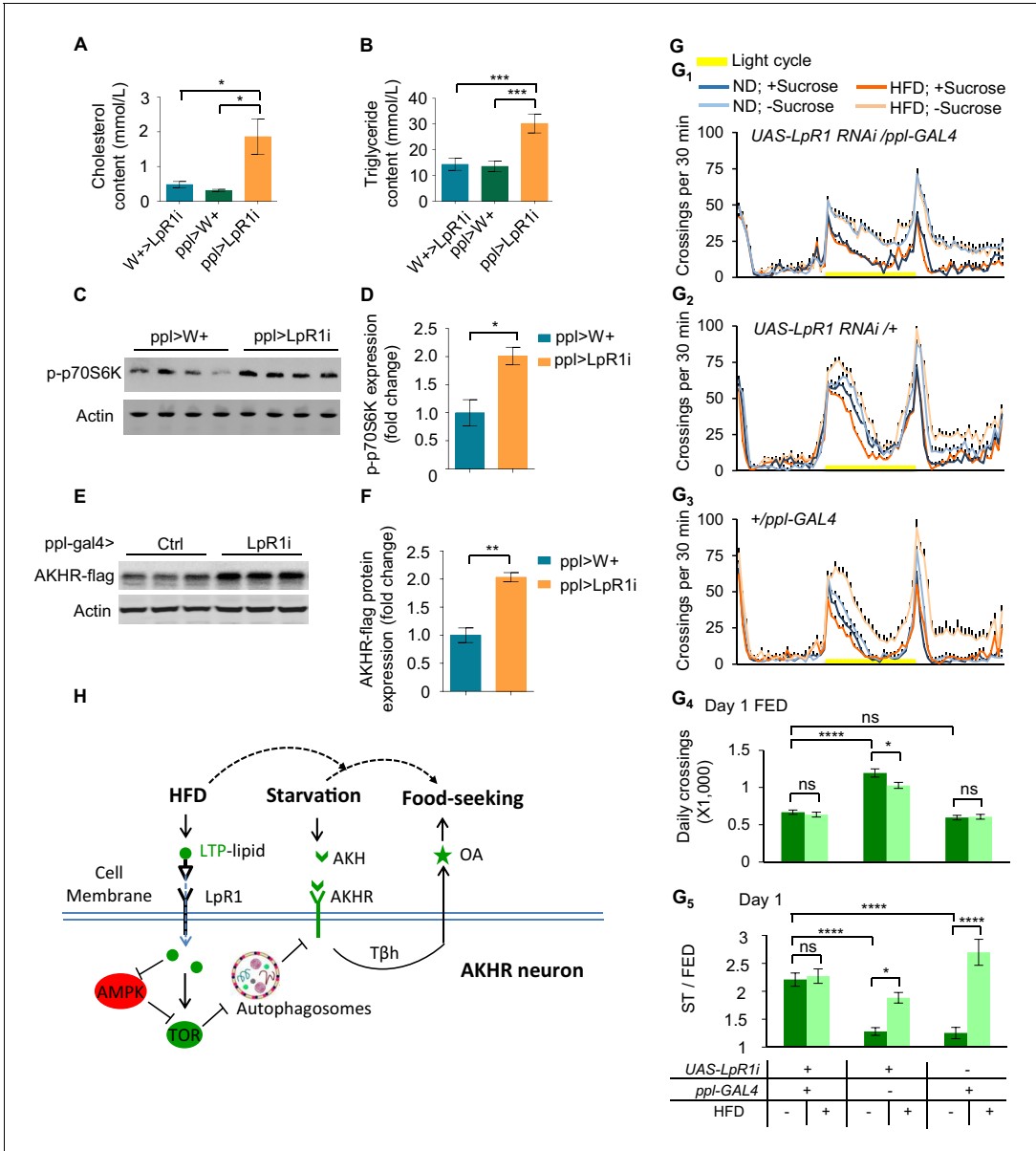

**Figure 8.** Diet-independent hyperlipidemia promotes the excitability of AKHR⁺ neurons and starvation-induced hyperactivity. (A–B) Average levels of cholesterol (A) and triglyceride (B) in the hemolymph of the indicated genotypes (n = 4–11 biological replicates, each containing 60 flies). (C–F) The levels of phosphorylated p70s6k (C–D) and AKHR-flag (E–F) from the head tissues of indicated genotypes (n = 4 biological replicates, each containing 25 flies). (G1-3) Midline crossing activity of indicated genotypes and diet treatments assayed in the presence or absence of 5% sucrose (n = 41–63). (G4) Average daily midline crossing activity of fed flies assayed in G1-G3. (G5) Starvation-induced hyperactivity of flies assayed in G1-G3. ns, p>0.05; *p<0.05; ***p<0.001; ****p<0.0001. Student's t-test, one-way and two-way ANOVA followed by post hoc test with Bonferroni correction were used for multiple comparisons when applicable. (H) A working model: We propose the following working model based on our results. AKHR protein levels in AKHR⁺ neurons are well maintained via protein expression and autophagic protein degradation. Under ND feeding, starvation induces the expression and release of AKH, which in turn activates AKHR⁺ neurons and hence the release of OA. OA signaling promotes starvation-induced hyperactivity in these starved flies. Upon HFD feeding, however, the regulation of starvation-induced hyperactivity is disrupted and greatly enhanced. Mechanistically, LTP delivers excess dietary lipids to AKHR⁺ neurons via its cognate receptor LpR1. Subsequently, neuronal autophagy is suppressed in AKHR⁺ neurons via AMPK-TOR signaling, which leads to AKHR accumulation. As a result, AKHR⁺ neurons become hyper sensitive to AKH and therefore starvation-induced hyperactivity is enhanced.

The online version of this article includes the following source data for figure 8:

**Source data 1.** Raw data of the behavioral experiments shown in *Figure 8*.

increased AKHR protein accumulation, the sensitivity of these neurons to AKH, and hence starvation-induced hyperactivity, all of which mimicked the effect of HFD feeding. Inhibiting TOR activity exerted an opposite effect. In addition, we also found that the effect of HFD on TOR signaling was mediated by AMPK signaling. These results altogether suggest that AMPK-TOR signaling in AKHR$^+$ neurons plays an important role in maintaining the homeostasis of these neurons and determining the responsiveness to HFD feeding. Similarly, rodent studies have shown that manipulating AMPK-TOR signaling results in the dysfunction of NPY/AgRP neurons as well as POMC neurons, which leads to abnormal food consumption and adiposity (*Kong et al., 2016*; *Yang et al., 2012*). It will be of interest to examine whether HFD also modulates AMPK-TOR signaling in these specific hypothalamic neurons.

We next sought to understand how AKHR$^+$ neurons detected HFD, or more specifically, excess lipid ingested by the flies. As an essential nutrient and important energy reserve, dietary lipids were transported via their carrier proteins, named lipoproteins, in the circulation system and regulated multiple cellular signaling pathways. Proteomic analysis helped us to identify that one lipoprotein LTP was enriched in flies' hemolymph after HFD feeding. Single-cell RNAseq of AKHR$^+$ neurons identified a number of lipoprotein receptors, especially LpR1, highly expressed in these neurons. Therefore, we proposed that AKHR$^+$ neurons might sense HFD feeding via LTP-LpR1 signaling. Evidently, we found that eliminating LpR1 in AKHR$^+$ neurons could protect flies from HFD, reducing AKHR accumulation and abolishing the effect of HFD to enhance starvation-induced hyperactivity. Conversely, eliminating LpR1 in the fat body, the major lipid reservoir of flies, created diet-independent hyperlipidemia and mimicked the effect of HFD feeding on flies' starvation-induced hyperactivity. Taken together, we propose a working model that upon HFD feeding, excess dietary lipids are transported by LTP in the hemolymph, which interacts with its cognate receptor LpR1 in OA$^+$AKHR$^+$ neurons. As a result, these neurons undergo a number of cellular signaling processes and eventually become hypersensitive to starvation (*Figure 8H*).

To summarize, our present study establishes a link between an unhealthy diet and abnormalities of food intake related behaviors in a model organism. We also decipher the underlying mechanism involving intracellular AMPK-TOR signaling, reduced neuronal autophagy, accumulation of a specific hormone receptor, and increased excitability of a small group of hunger-sensing neurons. Our study will shed crucial light on the pathological changes in the central nervous system upon metabolic challenges. Given that the central control of metabolism and food intake related behaviors are highly conserved across different species, it will be of importance to further examine whether similar mechanisms also mediate the effect of HFD feeding on food intake and metabolic diseases in mammals.

# Materials and methods

### Key resources table

| Reagent type (species) or resource | Designation | Source or reference | Identifiers | Additional information |
|---|---|---|---|---|
| Genetic reagent (*D. melanogaster*) | *UAS-TOR$^{TED}$* | Bloomington *Drosophila* Stock Center | Cat. #7013; RRID:BDSC_7013 | FlyBase symbol: P{w+mC=UAS-Tor.TED} |
| Genetic reagent (*D. melanogaster*) | *UAS-AMPK-DN* | Bloomington *Drosophila* Stock Center | Cat. #32112; RRID:BDSC_32112 | FlyBase symbol: P{w+mC=UAS-AMP Kalpha.K57A} |
| Genetic reagent (*D. melanogaster*) | *UAS-mCD8GFP* | Bloomington *Drosophila* Stock Center | Cat. #32186; RRID:BDSC_32186 | FlyBase symbol: P{y+t7.7 w+mC = 10 XUAS-IVS-mCD8::GFP} |
| Genetic reagent (*D. melanogaster*) | *UAS-mCD8GFP* | Bloomington *Drosophila* Stock Center | Cat. #32186; RRID:BDSC_32186 | FlyBase symbol: P{y+t7.7 w+mC = 10 XUAS-IVS-mCD8::GFP} |
| Cell line (include species here) | *ppl-GAL4* | Bloomington *Drosophila* Stock Center | Cat. #58768; RRID:BDSC_58768 | FlyBase symbol: P{w+mC = ppl-GAL4.P} |
| Genetic reagent (*D. melanogaster*) | *UAS-NaChBac* | Bloomington *Drosophila* Stock Center | Cat. # 9469; RRID:BDSC_9469 | FlyBase symbol: P{w+mC = UAS-NaChBac} |

*Continued on next page*

*Continued*

| Reagent type (species) or resource | Designation | Source or reference | Identifiers | Additional information |
|---|---|---|---|---|
| Genetic reagent (*D. melanogaster*) | *UAS-nSyb-AD-AKHR-BD* | Our laboratory PMID:27612383 | | Dr. Liming Wang (Zhejiang University) |
| Genetic reagent (*D. melanogaster*) | *UAS-nSyb-AD* | Our laboratory PMID:27612383 | | Dr. Liming Wang (Zhejiang University) |
| Genetic reagent (*D. melanogaster*) | *UAS- AKHR-BD* | Our laboratory PMID:27612383 | | Dr. Liming Wang (Zhejiang University) |
| Genetic reagent (*D. melanogaster*) | *UAS- AKHR$^{-/-}$* | Our laboratory PMID:27612383 | | Dr. Liming Wang (Zhejiang University) |
| Genetic reagent (*D. melanogaster*) | *UAS- kir2.1* | Our laboratory PMID:30209352 | | Dr. Liming Wang (Zhejiang University) |
| Genetic reagent (*D. melanogaster*) | *UAS-TSC1 RNAi* | Tsinghua Fly Center | Cat. #5074 | |
| Genetic reagent (*D. melanogaster*) | *UAS-ATG7 RNAi* | Tsinghua Fly Center | Cat. #2793 | |
| Genetic reagent (*D. melanogaster*) | *UAS-LpR1 RNAi* | Tsinghua Fly Center | Cat. #2568 | |
| Genetic reagent (*D. melanogaster*) | *UAS-ATG5 RNAi* | Laboratory of Dr. Chao Tong | | Dr. Chao Tong (Zhejiang University) |
| Genetic reagent (*D. melanogaster*) | *AKHR-FLAG* | This paper | | Dr. Liming Wang (Zhejiang University) |
| Cell line (*D. melanogaster*) | S2 | Thermofisher | Cat. #R69007 | |
| Antibody | Anti-phospho-Akt (Ser473) (Rabbit polyclonal) | Cell Signaling Technology | Cat. #9271; RRID:AB_329825 | WB (1:1000) |
| Antibody | Anti-phospho-AMPK (Thr172) (Rabbit polyclonal) | Cell Signaling Technology | Cat. #2535; RRID:AB_331250 | WB (1:1000) |
| Antibody | Anti-GFP (Rabbit polyclonal) | Cell Signaling Technology | Cat. #2555; RRID:AB_10692764 | WB (1:1000) |
| Antibody | Anti-FLAG (Rabbit polyclonal) | Cell Signaling Technology | Cat. #86861; RRID:AB_2800094 | WB (1:1000) |
| Antibody | Anti-HA (Rabbit polyclonal) | Cell Signaling Technology | Cat. #3724; RRID:AB_1549585 | WB (1:1000) |
| Antibody | Anti-HA (Mouse monoclonal) | Cell Signaling Technology | Cat. #2367, RRID:AB_10691311 | IF(1:200); WB (1:1000) |
| Antibody | Anti-actin (Rabbit polyclonal) | Sigma-Aldrich | Cat. #A5316; RRID:AB_476743 | WB (1:5000) |
| Antibody | Anti-AKH (Rabbit polyclonal) | biorbyt | Cat. #orb97730 | DOT BLOT (1:1000) |
| Antibody | Anti-Dilp2 (Rabbit polyclonal) | Laboratory of Dr. Zhefeng Gong | | DOT BLOT (1:1000) Dr. Zhefeng Gong (Zhejiang Universtiy) |
| Recombinant DNA reagent | AKHR-HA (plasmid) | This paper | | Dr. Liming Wang (Zhejiang University) |
| Recombinant DNA reagent | GFP-ATG8 (plasmid) | This paper | | Dr. Liming Wang (Zhejiang University) |
| Peptide, recombinant protein | AKH | PMID:2117437 | | pGlu-Leu-Thr-Phe-Ser -Pro-Asp-Trp-NH2 |
| Chemical compound, drug | MG132 | Beyotime Biotechnology | Cat. #S1748 | |
| Chemical compound, drug | AICAR | Beyotime Biotechnology | Cat. #S1516 | |

*Continued on next page*

*Continued*

| Reagent type (species) or resource | Designation | Source or reference | Identifiers | Additional information |
|---|---|---|---|---|
| Chemical compound, drug | MHY (MHY1485) | Sigma-Aldrich | Cat. #SML0810 | |
| Chemical compound, drug | dorsomorphin | Sigma-Aldrich | Cat. #P5499 | |
| Chemical compound, drug | Chloroquine (CQ) | Sigma-Aldrich | Cat. #C6628 | |
| Chemical compound, drug | Rapamycin | Sangon Biotech | Cat. #A606203 | |
| Software, algorithm | GraphPad Prism 6 | GraphPad Software | | www.graphpad.com |

## Flies

Flies were reared on standard fly food made of yeast, corn, and agar at 25°C and 60% humidity on a 12 hr light-12-hr dark cycle. Unless otherwise specified, virgin female flies were collected shortly after eclosion and placed in standard fly food (ND) vials (20 flies per vial) for 5–6 d prior to experiments. High fat diet (HFD)-fed flies were kept in vials with HFD medium containing 20% coconut oil and 80% ND for 5 d. HSD -fed flies were raised with ND plus 25% sucrose for 5 d. For rapamycin/dorsomorphin feeding, flies were raised on ND containing 400 μM rapamycin or 50 μM dorsomorphin for 5 d. *nSyb-AD, AKHR-BD, and AKHR$^{-/-}$* flies were described previously (*Yu et al., 2016*). *UAS-TSC1 RNAi* (#5074), *UAS-ATG7 RNAi* (#2793) and *UAS-LpR1 RNAi* (#2568) were obtained from the Tsinghua Fly Center. *UAS-TOR$^{TED}$* (#7013), *UAS-AMPK-DN* (#32112), *UAS-mCD8GFP* (#32186) and *ppl-GAL4* (#58768) were provided by the Bloomington *Drosophila* Stock Center at Indiana University. *UAS-ATG5 RNAi* flies were from Chao Tong (Zhejiang University). Fly strains were backcrossed for at least eight generations before tested in behavioral assays.

## Chemicals, plasmids, and antibodies

If not otherwise indicated, all chemicals were from Sigma. MG132 (10 μM; Beyotime Biotechnology) and CQ (100 μM) exposure was for 4 hr. Rapamycin (200 nM; Sangon Biotech), AICAR (500 μM; Beyotime Biotechnology), MHY (5 μM) and dorsomorphin (5 μM) were added to the medium for 12 hr. pAC5.1-flag and pAC5.1-HA vectors were kindly provided by Xiaohang Yang (Zhejiang University). AKHR-HA was made by cloning the cDNA of AKHR into a pAC5.1-HA vector using Sall and Notl. The cDNA of GFP-ATG8 was from Chao Tong (Zhejiang University) and cloned into a pAC5.1-HA vector using EcoRI and BamHI to generate HA-GFP-ATG8.

The following antibodies were used: rabbit polyclonal antibodies to p-AKT, p-AMPK, HA, Flag, GFP and mouse monoclonal antibody to HA (Cell Signaling Technology); mouse monoclonal antibody to actin (sigma); rabbit polyclonal antibody to AKH (biorbyt); rabbit polyclonal antibody to Dilp2 was kindly provided by Zhefeng Gong (Zhejiang University).

## Behavioral assays

DAMS-based locomotion assay was performed as described in our earlier report (*Yu et al., 2016*). Briefly, individual virgin female flies were introduced into 5 mm ×65 mm polycarbonate tubes (Trikinetics) after being lightly anesthetized. One end of these tubes was filled with medium containing 2% (wt/vol) agar ±5% (wt/vol) sucrose and the other end was blocked using cotton wool. These tubes were then inserted into DAMS monitors placing in fly incubators. The passage of flies through the middle of the tube was counted by an infrared beam through the midline of the tubes. The ratio of ST/FED was calculated by dividing the total crossing numbers collected from ZT0 to ZT24 in Day one and/or Day 2 of the locomotion assay under -sucrose vs. +sucrose conditions.

The MAFE assay was performed as described previously (*Qi et al., 2015*). Briefly, individual flies were gently aspirated and introduced into a 200 μL pipette tip. After exposing the proboscis by cutting the pipette tip, the flies were first sated with water and then presented with 100 mM sucrose containing 2% blue dye (McCormick) filled in a graduated capillary (VWR, #53432–604). Until the flies

became unresponsive to 10 serial food stimuli, food consumption was calculated according to the total volume of ingested food.

The FLIC assay was performed as described previously (*Qi et al., 2015*). Briefly, both feeding channels in the *Drosophila* Feeding Monitors (DFM) were filled with 5% sucrose. Individual flies were then gently aspirated into each feeding arena and their feeding activity was recorded for 24 hr. An electrical current generated by the physical contact between flies' proboscis and the liquid food was recorded by the DFMs. From the original report, electrical current more than 120 a.u. was considered as actual feeding, and the total feeding bouts and the duration of feeding time were calculated accordingly.

The BARCODE feeding was performed as described previously (*Park et al., 2018*). Briefly, fly food was mixed with one DNA oligomer (0.35 ng/ µl, 5'GGGCAGCAGGATAACTCGAATGTCTTAG TGCTAGAGGCTTGGGGCGTGTAAGTGTATCGAAGAAGTTCGTGTTAAACGCTTTGGAATGACTG TAATGTAG3'). After feeding for 1 hr, flies were washed to remove oligomer bound to the exterior and then homogenized for BARCODE qPCR. Forward qPCR Primer: 5' – CAGCAGGATAACTCGAA TGTCTTA – 3' Reverse qPCR Primer: 5' – CAGTCATTCCAAAGCGTTTAACA – 3'.

The video recording-based food seeking assay was performed as described previously (*Tian and Wang, 2018*). Briefly, individual flies were starved for 24 hr and then transferred into a behavioral chamber (84 mm ×84 mm) with a small food patch (5% sucrose) in the center. The positions and moving trajectories of invidivial flies were recorded by a camera placed on top and analyzed by a custom computer program.

## Calcium imaging

The flies were lightly anesthetized on ice, and brains were dissected in oxygen saturation (95% $O_2$, 5% $CO_2$) buffer extracellular saline solution [103 mM NaCl, 3 mM KCl, 5 mM N-Tris (hydroxymethyl) methyl-2-aminoethane-sulfonic acid, 10 mM trehalose, 8 mM glucose, 26 mM $NaHCO_3$, 1 mM $NaH_2PO_4$, 1.5 mM $CaCl_2$, and 4 mM $MgCl_2$, The pH was adjusted to 7.5 and osmolarity to 265 mOsm].

Dissected brains were adhered to a dish. Dorsal view is upward. Micropipettes (glass capillaries, Harvard Apparatus, 300092) with an opening diameter of approximately 10 µm were connected to a picospritzer III (Intracell). The tip of the pipette was positioned 50 µm to the cell body (*Figure 2I*). Air pressure on the picospritzer (III) was set to 3.3 psi, duration was 300 ms, the proper parameter required to eject solution from the pipette to induce the calcium response in AKHR promoter driven calcium indicator GCamp6m. Electrode was perfused by microloader. Alexa fluor 568 was mixed in the AKH solution (3.16 mM) to verify the eject area could cover the target AKHR neuron's cell body (*Figure 2I*).

Images were acquired on Nikon's Eclipse FN1 confocal microscopes with 40×/0.8 objectives. Time series were acquired at 1 Hz with 256 × 256 pixels and 12 bit, a resolution of 0.83 µm/ pixel, controlled by NIS elements. Laser power density on the scanned area of laser wavelength 488 nm was lower than 0.3 mW/ mm$^2$, and 561 nm was 10 mW/mm$^2$.

Total image acquired duration is 4 min, first 1 min set to pre-puff drug time followed by 300 ms drug delivered. Image stacks were subsequently analyzed by Fiji (*Schindelin et al., 2012*; http://fiji. sc). ROIs were manually drawn around AKHR$^+$ neuron cell body, and the intensity of 30 frames before puff time average to be F0, individual ΔF/F0 response traces were extracted, and the peak ΔF/F0 was obtained for significance analysis.

## Immunofluorescence staining

For immunostaining, *Drosophila* S2 cells were fixed in 4% formaldehyde. After washing thrice in PBS, cells were blocked in PBS/FBS (PBS, pH 7.4, containing 10% FBS). The cells were then incubated with appropriate primary and secondary antibodies in PBS/FBS with 0.1% saponin.

## RNA-seq and analysis

Total RNA from fly heads was extracted from 5-day-old female flies using the Trizol reagent (Invitrogen, USA), subjected to poly(A) mRNA isolation, cDNA synthesization, library preparation (NEBNext Ultra II DNA Library Prep Kit, NEB), and sequencing (Illumina Hiseq2500/4000 platform). Sequence data were subsequently mapped to *Drosophila* genome and uniquely mapped reads were collected

for further analysis. Gene expression was calculated by the FPKM (Fragments Per Kilobase Of Exon Per Million Fragments Mapped). The genes with p-value less 0.05 and fold change more than two were considered as the differentially expressed gene. Functional annotation of detected differentially expressed genes was assessed by publicly available databases including Gene Ontology (GO) (http://www.geneontology.org) and Kyoto Encyclopedia of Genes and Genomes (KEGG) (http://www.genome.jp/kegg). The RNAseq data were deposited in GEO database under the accession code GSE129602.

## Single-cell RNAseq

As described in our previously report (*Yu et al., 2016*), individual AKHR[+] cells (marked by GFP expression) were picked from dissected fly brain under a fluorescence microscope using a glass micropipette pulled from thick-walled borosilicate capillaries (BF120-69-10, Sutter Instruments). Separated cells were transferred to lysate buffer immediately, followed by reverse transcription and cDNA amplification (SMARTer Ultra Low RNA Kit for Sequencing, Clonetech). The amplified cDNA underwent library preparation using the NEBNext Ultra II DNALibrar kit and subject to sequencing by Illumina Hiseq 2500 platform. The sequenced raw data were purified to remove low-quality reads, adaptor sequences and amplification primer before mapping to *Drosophila* genome. Only mapped reads were selected for further analysis. FPKM (Fragments Per Kilobase Of Exon Per Million Fragments Mapped) was used to quantify gene expression. The RNAseq data were deposited in GEO database under the accession code GSE129601.

## Measurements of CO2 production

The CO2 production was performed as described previously (*Takeuchi et al., 2009*). 1 ml plastic syringes were used for the respirometers after filling a small amount of CO2 absorbent (Soda lime, Sigma) between two pieces of sponge in the body of the syringe. A 5 µl fine graduated capillary connected to the plastic adaptor attached to the syringe. Five flies were gently aspirated into the syringe body and plunger inserted to form a closed chamber. After being kept on the flat surface at 25°C for 15 min to equilibrate, a little ink was placed at the end of the micropipet. The CO2 production was calculated based on the rate of movement of the ink.

## Triglyceride and cholesterol measurement

For whole fly bodies, single fly was anesthetized and transferred to 400 µL of 0.5% PBST, subject to homogenization (Tissuelyser 24, Qiagen, USA) and incubation at 92°C for 10 min. After centrifugation, the supernatant was measured by Triglyceride Quantification Colorimetric/Fluorometric Kit (BioVision, USA).

For fly hemolymph, extracted hemolymph was directly measured by Triglyceride Assay or Cholesterol Assay Kit (Nanjing Jiancheng Bioengineering Institute, China).

## Hemolymph extraction

40 flies were decapitated and transferred to a punctured 0.5 ml tube. The tube was then placed into 1.5 ml eppendorf tube, subjected to centrifugation for 5 min at $2500 \times g$ at 4°C. Collected hemolymph in 1.5 ml eppendorf tube was used for further analysis.

## LC-MSms/MS

LC-MS/MS analysis was performed on HPLC chromatography system named Thermo Fisher Easy-nLC 1000 equipped with a C18 colume (1.8 mm, 0.15 × 1.00 mm). The MS/MS data were searched with MaxQuant (version 1.5.2.8). The following parameters were used: (i) enzyme: trypsin; (ii) fixed modification: carbamidomethyl (C); (iii) variable modifications: oxidation (M) and deamidation (NQ); (iv) mass tolerance for precursor ions: 20 ppm; (v) mass tolerance for fragment ions: 4.5 ppm; (vi) peptide charge: 1+, 2+ and 3+; (vii) instrument: ESI-LTQ; (viii) allowing up to two missed cleavage. (ix) minimal peptide length: six amino acid. (x) the false discovery rate (FDR) for peptide and protein identifications: 0.01. The intensity for each protein was obtained from three technical repeats. The global analysis was carried by in-house program. The criteria for the interesting proteins were: the ratio of intensity between the experimental and control samples larger than two and p-value<0.05 was considered to be statistically significant. The raw data was shown in *Figure 7—source data 1*.

## Western blotting and dot blotting

For western blot, total protein was extracted from fly heads or cultured S2 cells. Samples were denatured, separated by SDS-PAGE, and transferred to a polyvinylidene difluoride membrane. After being blocked in TBST containing 5% milk, the membrane was incubated with the specific primary antibody followed by HRP-conjugated goat-anti-rabbit or goat anti-mouse antibody. The specific bands were detected by an ECL western blotting detection system (Bio-rad, USA).

For dot blot, hemolymph was extracted and spotted onto nitrocellulose membrane and let dry for 20 min. After washing, blocking, incubation with the corresponding primary and secondary antibodies, the membranes were also analyzed by an ECL western blotting detection system.

## Quantitative RT-PCR

Total RNA from fly heads was isolated from 5-day-old female flies with TRIzol reagent (Invitrogen, USA). Reverse transcription of RNA to cDNA was achieved using 2 μg of total RNA with random hexamer primers using RT-PCR SuperMix Kit (TransGen Biotech, China). Q-PCR was performed with SYBR premix Ex TaqTM (Takara, China) on CFX96 Touch Real-Time PCR Detection System (Bio-Rad, USA). The primers used are presented as follows: DILP2 F: 5′-GCCTTTGTCCTTCATCTCG-3′, DILP2 R: 5′-CCATACTCAGCACCTCGTTG-3′; AKH F: 5′- ATCCCAAGAGCGAAGTCC −3′, AKH R: 5′- CCTGAGATTGCACGAAGC −3′; AKHR F: 5′- ACTGCTACGGAGCCATTT −3′, AKHR R: 5′- TGTCCAGCCAGTACCACA −3′; InR F: 5′- GCAGCAATGAATGCGACGAT −3′, InR R: 5′- CCTGCGTCGCTTGTTGAAAA −3′; GAPDH F: 5′-GAATCCTGGGCTACACCG-3′, GAPDH R: 5′-CTTATCGTTCAGAATGC-3′.

## Cell lines

Schneider's Line S2 (S2) cells (Thermo Fisher, R69007) were maintained in the lab and were all mycoplasma-free. The cells were grown in Schneider's Insect Medium (sigma) supplemented with 10% fetal bovine serum (FBS) at 25°C. Transfections were performed using Lipofectamine 3000 according to the manufacturer's instructions (Invitrogen). Cells were analyzed 24–48 hr after transfection.

## Statistical analysis

Data presented in this study were verified for normal distribution by D'Agostino–Pearson omnibus test. Student's t test, one-way ANOVA and two-way ANOVA (for comparisons among three or more groups and comparisons with more than one variant) were used. The post hoc test with Bonferroni correction was performed for multiple comparisons following ANOVA. The raw data for all behavioral assays were shown in source data files associated with Figures.

## Acknowledgements

We thank members of the Neuroscience Pioneer Club for insightful discussions throughout the course of the study. We thank all Wang Lab members and Tingzhang Wang and Chen Pan (Zhejiang University) for helpful discussions and technical assistance. We thank Fuchou Tang (Peking University) for help with the single cell analysis. We thank Yeguang Chen (Tsinghua University) and Wei Liu (Zhejiang University) for helpful comments. Ye Wu provides administrative support in the laboratory. This study was funded by National Key Research and Development Program of China (2019YFA0802400 and 2019YFA0801900 for LW), the National Natural Science Foundation of China (No. 31522026 for LW and No. 31800883 for RH), the Thousand Young Talents Plan (LW), and the Fundamental Research Funds for the Central Universities (No. 2016QN81010, No. 2015XZZX004-33 for LW, and 2019CDYGYB009 for RH).

## Additional information

### Funding

| Funder | Grant reference number | Author |
| --- | --- | --- |
| National Natural Science Foundation of China | 31522026 | Liming Wang |

| National Natural Science Foundation of China | 31800883 | Rui Huang |
|---|---|---|
| National Key Research and Development Program of China | 2019YFA0802400 | Liming Wang |
| National Key Research and Development Program of China | 2019YFA0801900 | Liming Wang |
| Thousand Talents Plan | | Liming Wang |
| Fundamental Research Funds for the Central Universities | 2016QN81010 | Liming Wang |
| Fundamental Research Funds for the Central Universities | 2015XZZX004-33 | Liming Wang |
| Fundamental Research Funds for the Central Universities | 2019CDYGYB009 | Rui Huang |

The funders had no role in study design, data collection and interpretation, or the decision to submit the work for publication.

### Author contributions

Rui Huang, Conceptualization, Resources, Funding acquisition, Validation, Investigation, Visualization, Methodology, Writing - original draft, Writing - review and editing; Tingting Song, Validation, Investigation, Visualization, Methodology, Writing - original draft, Writing - review and editing; Haifeng Su, Investigation, Methodology; Zeliang Lai, Wusa Qin, Xuan Dong, Investigation; Yinjun Tian, Resources, Investigation; Liming Wang, Conceptualization, Resources, Supervision, Funding acquisition, Validation, Investigation, Writing - original draft, Project administration, Writing - review and editing

### Author ORCIDs

Rui Huang (iD) https://orcid.org/0000-0003-4656-1682
Liming Wang (iD) https://orcid.org/0000-0002-7256-8776

### Decision letter and Author response

Decision letter https://doi.org/10.7554/eLife.53103.sa1
Author response https://doi.org/10.7554/eLife.53103.sa2

## Additional files

### Supplementary files

• Transparent reporting form

### Data availability

Sequencing data have been deposited in GEO under accession codes GSE129601 and GSE129602. All raw data for mass spectrometry and behavioural experiments are included as source data files.

The following datasets were generated:

| Author(s) | Year | Dataset title | Dataset URL | Database and Identifier |
|---|---|---|---|---|
| Huang R, Wang L | 2019 | High-fat diet enhances food-seeking behavior via sensitizing hunger-sensing neurons in Drosophila I | http://www.ncbi.nlm.nih.gov/geo/query/acc.cgi?acc=GSE129601 | NCBI Gene Expression Omnibus, GSE129601 |
| Huang R, Wang L | 2019 | High-fat diet enhances food-seeking behavior via sensitizing hunger-sensing neurons in Drosophila II | http://www.ncbi.nlm.nih.gov/geo/query/acc.cgi?acc=GSE129602 | NCBI Gene Expression Omnibus, GSE129602 |

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
