## [Decision Letter]

**Acceptance summary:**

This article provides a great deal of new information about the mechanism by which a high-fat diet enhances hyperactivity in starved flies. Many steps of the pathway are investigated, using many experimental approaches. The pathway that emerges is analogous to corresponding pathways in mammals at several steps, but some steps have not been described in mammals and may provide a useful guide to those studying mammalian metabolism. Overall the work represents a valuable contribution to the field.

**Decision letter after peer review:**

Thank you for submitting your article "High-fat diet enhances starvation-induced hyperactivity via sensitizing hunger-sensing neurons in *Drosophila*" for consideration by *eLife*. Your article has been reviewed by two peer reviewers, and the evaluation has been overseen by K VijayRaghavan as the Reviewing Editor and Senior Editor. The reviewers have opted to remain anonymous.

The reviewers have discussed the reviews with one another and the Reviewing Editor has drafted this decision to help you prepare a revised submission.

Summary:

This article provides a great deal of new information about the mechanism by which a high-fat diet enhances hyperactivity in starved flies. Steps of the pathway are investigated, using many experimental approaches. The authors found the effect of high-fat diet (HFD) on starvation-induced hyperactivity in flies, and further dissected the molecular mechanisms of it using various approaches such as behavioral analysis, calcium imaging, pharmacological studies, single-cell sequencing, etc. They identified an HFD-AMPK-TOR-autophagy-AKHR candidate pathway underlying HFD-enhanced starvation-induced hyperactivity. The pathway that emerges is analogous to corresponding pathways in mammals at several steps, but some steps have not been described in mammals and may provide a useful guide to those studying mammalian metabolism. Overall the work represents a valuable contribution to the field. However, several improvements on the experiments and manuscript could be made to make this study more solid and better interpreted. The concerns are listed below.

Essential revisions:

1) The behavioral output that is measured here is hyperactivity, not food-seeking. An earlier paper by these authors (Yang et al., 2015) provided some evidence that this hyperactivity is related to food-seeking. However, the present paper shows that a high-fat diet (HFD) did not alter flies' food consumption or energy expenditure. The significance of hyperactivity needs more discussion.

2) The major interest of this study is to dissect the mechanisms underlying HFD-enhanced starvation-induced hyperactivity. In this context, there are two variables for this behavioral output: HFD comparing with ND (normal diet), and starvation (ST) comparing with feeding (FED). The authors used Daily crossings (x1,000) to measure the FED activity, and used ST/FED to measure starvation-induced hyperactivity. As the most important measurement, a detailed description of the calculation method of ST/FED should be made in the manuscript. Is it referring to the ratio of daily crossings between ST and FED, under ZT0-ZT24?

3) The authors show that flies exhibit comparable feeding activity on HFD and a ND. If the HFD is more calorically dense, does this mean that their caloric intake is higher? If so, is this a confounding variable?

4) In Figure 2, the authors suggest that HFD enhances starvation-induced hyperactivity through AKHR based on two results: (1) AKHR protein expression was increased in HFD flies at FED status; (2) AKHR neurons are more sensitive to AKH in HFD flies.

More evidences are needed to make this statement more convincing. For example:

a) What's the AKHR protein level under ST in HFD and ND flies comparing with FED?

b) What's the phenotype of *AKHR^-/-^* flies fed with HFD under starvation?

c) What's the effect of silencing/activation AKHR neurons on this HFD enhanced starvation-induced hyperactivity?

5) Much of the activity data is presented as a ratio of Starved/Fed. To help interpret this ratio it would be useful to see the Starved values separately.

6) In Figure 4F, the blockage of AKHR>ATG-5 RNAi on the ST/FED change between ND and FED seems to be majorly caused by a huge and significant increase of ST/FED at ND comparing with the control groups, but not a decrease of ST/FED at HFD, which makes the conclusion of HFD enhancing ST/FED through autophagy less convincing. Is autophagy blockage affecting starvation-induced hyperactivity under ND rather than affecting starvation-induced hyperactivity under HFD? And the way the statistical difference being labeled in this kind of graphs is also confusing: are the authors comparing the 1st bar with the 3rd bar, or comparing the red group with the blue group?

7) Is the pathway AMPK-TOR-autophagy-AKHR an HFD specific pathway or a ND/HFD common pathway? As shown in Figure 6F, TOR inhibition also significantly decreased ST/FED under ND. Is HFD enhancing a ND starvation-induced hyperactivity pathway?

8) The statistical difference in Figure 1I and Figure 7D is similar, but the interpretation is quite different. In Figure 1I, the difference between ND and HFD is neglected, described as "One possible explanation was that HFD might decrease flies' energy storage and/or increase flies' energy expenditure, reducing their resistance to starvation (Figure 1C). It seemed unlikely, since flies' body weight was not affected by HFD and their lipid storage was actually slightly elevated by HFD (Figure 1H and I).", while in Figure 7D, the difference is confirmed, described as "Meanwhile, both triglycerides and cholesterol were enriched in the hemolymph of HFD-fed flies (Figure 7C-D). Therefore, it was possible that LTP was the major carrier for excess circulating lipids after HFD feeding." This is kind of selective interpretation.

9) Subsection “HFD enhanced AKHR accumulation by suppressing neuronal autophagy”: the body of the text should explain why the ratio of GFP-ATG8 I and II is a measure of autophagy, and a reference should be provided.

10) Dosage of AKH in Ca^2+^ imaging experiments: Figure 2K shows that HFD-fed flies exhibited more robust calcium transients than ND-fed flies when stimulated with AKH, but the experiment shows doses only up to 31.6 μM. In subsequent experiments, such as Figure 4A, it seems that 3.16 mM was used. This needs to be commented on. Also, the line graphs in Figure 2K are misleading – there is no 3.16 μM dose so the x-axis is not uniform in scale.

---

## [Author Response]

Essential revisions:1) The behavioral output that is measured here is hyperactivity, not food-seeking. An earlier paper by these authors (Yang et al., 2015) provided some evidence that this hyperactivity is related to food-seeking. However, the present paper shows that a high-fat diet (HFD) did not alter flies' food consumption or energy expenditure. The significance of hyperactivity needs more discussion.

As the reviewer pointed out, our previous work (Yang et al., 2015) showed that starvation-induced hyperactivity was a reliable (yet indirect) measure of food-seeking behavior in fruit flies. Besides showing that HFD-feeding enhanced starvation-induced hyperactivity (Figure 1), we have now added more direct measure of food-seeking behavior in the manuscript. As shown in Figure 1—figure supplement 3, starved flies were allowed to explore in a behavioral chamber with the presence of food sources. HFD-fed flies exhibited higher walking speed during the exploration period, and shorter latency to locate and occupy the food sources. These data further support the notion that HFD-feeding enhanced food-seeking behavior. We have also added more discussions in the main text (subsection “HFD specifically enhanced starvation-induced hyperactivity”, third paragraph).

We have also shown in our previous studies (Yang et al., 2015; Yu et al., 2016) that starvation-induced hyperactivity and starvation-induced increase in food consumption were regulated independently, aka manipulations that altered starvation-induced hyperactivity did not affect food consumption. Therefore, our current results that HFD feeding enhanced hyperactivity without altering food consumption are in line with our previous work. We have added more discussions in the manuscript (see the fifth paragraph of the aforementioned subsection).

2) The major interest of this study is to dissect the mechanisms underlying HFD-enhanced starvation-induced hyperactivity. In this context, there are two variables for this behavioral output: HFD comparing with ND (normal diet), and starvation (ST) comparing with feeding (FED). The authors used Daily crossings (x1,000) to measure the FED activity, and used ST/FED to measure starvation-induced hyperactivity. As the most important measurement, a detailed description of the calculation method of ST/FED should be made in the manuscript. Is it referring to the ratio of daily crossings between ST and FED, under ZT0-ZT24?

As the reviewers correctly pointed out, the ratio of ST/FED was calculated as the ratio between daily crossings (ZT0-ZT24) under ST and FED conditions. We have added the detailed description in the manuscript (subsection “Behavioral assays”).

3) The authors show that flies exhibit comparable feeding activity on HFD and a ND. If the HFD is more calorically dense, does this mean that their caloric intake is higher? If so, is this a confounding variable?

Indeed, HFD-fed flies probably had higher caloric uptake than ND-fed flies when fed ad libitum, which was also consistent with their higher triglyceride storage (Figure 1M). However, we did not consider this a confounding variable.

In particular, HFD-fed flies, despite their likely higher caloric update and higher triglyceride storage, exhibited increased (rather than decreased) response to starvation (Figure 1A-B, F). This seemingly counterintuitive result led to the suggestion that these flies exhibited enhanced starvation-induced hyperactivity *not* because of their lack of energy storage, but rather because of their hypersensitivity to hunger signals. The majority of our current study was built on and was to confirm this hypothesis (Figures 2-8).

We have added more discussions to clarify this point (subsection “HFD specifically enhanced starvation-induced hyperactivity”).

4) In Figure 2, the authors suggest that HFD enhances starvation-induced hyperactivity through AKHR based on two results: (1) AKHR protein expression was increased in HFD flies at FED status; (2) AKHR neurons are more sensitive to AKH in HFD flies.More evidences are needed to make this statement more convincing. For example:a) What's the AKHR protein level under ST in HFD and ND flies comparing with FED?b) What's the phenotype of AKHR^-/-^ flies fed with HFD under starvation?c) What's the effect of silencing/activation AKHR neurons on this HFD enhanced starvation-induced hyperactivity?

As suggested by the reviewer, we have added a number of new results to support our statement:

1) We have shown that consistent with the results from FED condition (Figure 2G-H), under Starved condition, HFD feeding also enhanced AKHR expression compared to ND. (Figure 2—figure supplement 1).

2) We have shown that *AKHR^-/-^* mutant flies exhibited no change in hyperactivity upon HFD-feeding (Figure 4—figure supplement 1).

3) We have shown that silencing of AKHR^+^ neurons by the expression of Kir2.1 also blocked the effect of HFD-feeding to enhance starvation-induced hyperactivity (Figure 4—figure supplement 2).

4) We have shown that activation of AKHR^+^ neurons by the expression of NaChBac enhanced starvation-induced hyperactivity under ND feeding (Figure 4—figure supplement 3).

Taken together, these newly added data further support the idea that AKHR expression and the neural activity of AKHR^+^ neurons are critical for the effect of HFD feeding to enhance starvation-induced hyperactivity. We have added the relevant discussions in the last paragraph of the subsection “HFD increased the sensitivity of hunger-sensing, AKHR+ neurons”.

5) Much of the activity data is presented as a ratio of Starved/Fed. To help interpret this ratio it would be useful to see the Starved values separately.

We have added the Starved values in Figure 1 (and Figure 1—figure supplements 1-5). However, we feel that adding Starved values into all figures (esp. those with already complicated genetic manipulations) would be making the figures a lot busier and harder to interpret. Furthermore, the Starved values alone do not offer much new insight since they are determined by both the baseline locomotion level (FED values) as well as the increase in locomotion upon starvation (ST/FED ratios), which are both already shown. Alternatively, we have added all the Starved values (and plots) into source data files for whom interested.

6) In Figure 4F, the blockage of AKHR>ATG-5 RNAi on the ST/FED change between ND and FED seems to be majorly caused by a huge and significant increase of ST/FED at ND comparing with the control groups, but not a decrease of ST/FED at HFD, which makes the conclusion of HFD enhancing ST/FED through autophagy less convincing. Is autophagy blockage affecting starvation-induced hyperactivity under ND rather than affecting starvation-induced hyperactivity under HFD? And the way the statistical difference being labeled in this kind of graphs is also confusing: are the authors comparing the 1st bar with the 3rd bar, or comparing the red group with the blue group?

Indeed, as the reviewer suggested, ATG5 KD led to significantly enhanced starvation-induced hyperactivity under ND (Figure 4F). But this is exactly as expected, since we proposed that HFD-feeding enhanced starvation-induced hyperactivity via suppressing autophagy and increasing AKHR levels.

Under ND feeding, ATG5 KD in AKHR^+^ neurons led to increased AKHR expression and enhanced neuronal sensitivity to AKH, mimicking the effect of HFD-feeding (Figure 3H-I and Figure 4A-B). Consistently, under ND-feeding, ATG5 KD flies exhibited enhanced starvation-induced hyperactivity (Figure 4F, dark bars), also mimicking HFD-feeding conditions (Figure 4F, dark vs. light bars for ATG5 KD flies).

Meanwhile, since HFD feeding has already suppressed autophagy to a low level (Figure 3F-G), further suppressing autophagy under HFD feeding did not exert any further increase in starvation-induced hyperactivity (Figure 4F, light bars).

We have added more clarifications in the Results.

We have modified the figures so that the statistical differences were indicated more clearly. Briefly, two short vertical lines indicate the two groups of data for statistical comparisons.

7) Is the pathway AMPK-TOR-autophagy-AKHR an HFD specific pathway or a ND/HFD common pathway? As shown in Figure 6F, TOR inhibition also significantly decreased ST/FED under ND. Is HFD enhancing a ND starvation-induced hyperactivity pathway?

The data shown in Figures 3-6 suggest that AMPK-TOR-autophagy-AKHR signaling is involved in starvation-induced hyperactivity under both ND and HFD conditions. Under HFD condition, this pathway is further modulated (AMPK down, TOR up, autophagy down, AKHR up) in a way that enhances starvation-induced hyperactivity.

As a result, as shown in Figure 6D, suppressing TOR signaling led to reduced starvation-induced hyperactivity under ND condition, and also the diminished behavioral response to HFD feeding.

We have added more clarifications in the manuscript. “These data suggest that TOR signaling is required for starvation-induced hyperactivity as well as the effect of HFD to further enhance this behavior”.

8) The statistical difference in Figure 1I and Figure 7D is similar, but the interpretation is quite different. In Figure 1I, the difference between ND and HFD is neglected, described as "One possible explanation was that HFD might decrease flies' energy storage and/or increase flies' energy expenditure, reducing their resistance to starvation (Figure 1C). It seemed unlikely, since flies' body weight was not affected by HFD and their lipid storage was actually slightly elevated by HFD (Figure 1H and I).", while in Figure 7D, the difference is confirmed, described as "Meanwhile, both triglycerides and cholesterol were enriched in the hemolymph of HFD-fed flies (Figure 7C-D). Therefore, it was possible that LTP was the major carrier for excess circulating lipids after HFD feeding." This is kind of selective interpretation.

We thank the reviewers for point out this problem and we have revised the interpretation (esp. for Figure 1M, Results), so that our interpretations for elevated TAG levels are consistent.

We would like to emphasize that in both experiments, the elevated triglyceride storage were supporting (not against) our conclusions.

In Figure 1M, the elevated triglyceride storage suggests that the enhanced starvation-induced hyperactivity upon HFD feeding is not a result of decreased energy supply, but rather a result of increased hunger sensitivity (which was proven right in the subsequent analysis). In Figure 7C-D, the elevated triglyceride and cholesterol levels in the hemolymph suggest that there should be some lipoprotein carrier for them (which was consistent with the elevated expression of LTP in the hemolymph).

9) Subsection “HFD enhanced AKHR accumulation by suppressing neuronal autophagy”: the body of the text should explain why the ratio of GFP-ATG8 I and II is a measure of autophagy, and a reference should be provided.

We have added more discussions and related references in the manuscript (subsection “HFD enhanced AKHR accumulation by suppressing neuronal autophagy”). Briefly, ATG8-I is the soluble form of ATG8 whereas ATG-II is the membrane-bound form. Upon the induction of autophagy, ATG8 is recruited to the membrane of newly formed autophagosomes. Therefore, an increased ratio of II/I indicates the induction of autophagy.

10) Dosage of AKH in Ca2+ imaging experiments: Figure 2K shows that HFD-fed flies exhibited more robust calcium transients than ND-fed flies when stimulated with AKH, but the experiment shows doses only up to 31.6 μM. In subsequent experiments, such as Figure 4A, it seems that 3.16 mM was used. This needs to be commented on. Also, the line graphs in Figure 2K are misleading – there is no 3.16 μM dose so the x-axis is not uniform in scale.

In the last revised manuscript, we have revised the figures to show Ca^2+^ imaging results of 316 nM AKH (Figure 4A-B, Figure 5G-H). We have added the information in the figure legends.

We have revised Figure 2K according to the reviewer’s suggestion.